# How microscopic epistasis and clonal interference shape the fitness trajectory in a spin glass model of microbial long-term evolution

**Nicholas M Boffi[1]\*, Yipei Guo[2], Chris H Rycroft[3,4], Ariel Amir[5,6]**

[1]Courant Institute of Mathematical Sciences, New York University, New York, United States; [2]Janelia Research Campus, Ashburn, United States; [3]Department of Mathematics, University of Wisconsin–Madison, Madison, United States; [4]Mathematics Group, Lawrence Berkeley National Laboratory, Berkeley, United States; [5]Weizmann Institute of Science, Rehovot, Israel; [6]John A. Paulson School of Engineering and Applied Sciences, Harvard University, Cambridge, United States

**\*For correspondence:**
boffi@cims.nyu.edu

**Abstract** The adaptive dynamics of evolving microbial populations takes place on a complex fitness landscape generated by epistatic interactions. The population generically consists of multiple competing strains, a phenomenon known as clonal interference. Microscopic epistasis and clonal interference are central aspects of evolution in microbes, but their combined effects on the functional form of the population's mean fitness are poorly understood. Here, we develop a computational method that resolves the full microscopic complexity of a simulated evolving population subject to a standard serial dilution protocol. Through extensive numerical experimentation, we find that stronger microscopic epistasis gives rise to fitness trajectories with slower growth independent of the number of competing strains, which we quantify with power-law fits and understand mechanistically via a random walk model that neglects dynamical correlations between genes. We show that increasing the level of clonal interference leads to fitness trajectories with faster growth (in functional form) without microscopic epistasis, but leaves the rate of growth invariant when epistasis is sufficiently strong, indicating that the role of clonal interference depends intimately on the underlying fitness landscape. The simulation package for this work may be found at https://github.com/nmboffi/spin_glass_evodyn.

## eLife assessment

This **important** study describes a high performance computational approach to interrogate how microscopic epistasis and clonal interference affect evolutionary dynamics in a spin glass model of microbial evolution. The study offers several insights that can aid in our understanding of the forces that operate in adaptive evolution. The evidence provided is **compelling**, with its rigorous use of models and analytical descriptions of how these forces manifest in evolution.

## Introduction

Laboratory evolution experiments have demonstrated the widespread prevalence of microscopic epistasis, the tendency for the phenotype associated with a mutation to depend on the background genotype in which it emerged (*Khan et al., 2011*; *Chou et al., 2011*; *Wang et al., 2013*; *Good and Desai, 2015*; *Bakerlee et al., 2022*; *Kryazhimskiy et al., 2014*; *DiazColunga et al., 2023*). Basic

evolutionary theory indicates that for moderate mutation rates the overall population will consist of many competing strains, because additional mutations can emerge before an existing mutation has time to fix in the culture (*Gerrish and Lenski, 1998*; *Desai and Fisher, 2007*; *de Visser and Rozen, 2006*; *Park and Krug, 2007*). This clonal interference is consistently observed in laboratory experiments such as Lenski's long-term evolution experiment (LTEE; *Fogle et al., 2008*; *Lenski et al., 1991*; *Lenski and Travisano, 1994*; *Lenski, 2017*; *Wiser et al., 2013*). Yet, despite the agreed-upon ubiquity of both aspects of evolution, there are few quantitative predictions for how they affect some of the most common experimental outputs, such as the mean fitness of the population. The central difficulty arises from the need to treat both the population and the genome at the microscopic level, which requires sophisticated analytical tools or high-resolution experiments.

The traditional approach in evolutionary theory is to make use of assumptions and statistical model classes that sidestep these complexities. Most theoretical studies of long-term adaptation take place in the strong selection weak mutation (SSWM) limit (*Gillespie, 1983*; *Gillespie, 1984*; *Gillespie, 1991*; *Orr, 2002*; *Good et al., 2012*), where the time for a beneficial mutation event to occur is large in comparison to the time for a typical beneficial mutation to fix in the population. While convenient due to analytical simplifications, this limit neglects clonal interference by ensuring that the culture consists of a single dominant strain at almost all times, and is hence known to be invalid for populations under standard laboratory conditions (*de Visser and Rozen, 2006*). As an unfortunate by-product, even if the predictions of a model are found to match experimental data, it is not clear how the addition of clonal interference will change the results.

To avoid the analytical and computational challenges associated with modeling the genome at microscopic granularity, significant theoretical effort has been spent studying macroscopic "rugged" or uncorrelated models (*Kauffman and Levin, 1987*; *Kauffman and Weinberger, 1989*; *Wilke, 2004*; *Macken and Perelson, 1989*; *Flyvbjerg and Lautrup, 1992*; *Kingman, 1978*), which posit the fitness of a mutant to be drawn at random from a fixed distribution (*Park and Krug, 2008*). More recently, macroscopic "fitness-parameterized" models, which assume the distribution of fitness effects (DFE) depends only on the fitness of the parent, have garnered interest as a way to model correlations in the fitness landscape (*Kryazhimskiy et al., 2009*). Although macroscopic models have provided significant insight into evolutionary dynamics, both classes exhibit serious disadvantages. Rugged landscapes make predictions that are known to violate experimental measurements, such as the typical length of an adaptive walk (*Orr, 2006*). Fitness-parameterized models can help correct some of these issues, but to do so they require an assumption about how the DFE depends on fitness, which is typically unknown because of its experimental intractability.

One approach that corrects the deficiencies of macroscopic models is the use of microscopic models, which treat the genome as a sequence of loci each with a binary label indicating the presence or absence of a mutation. Perhaps the most well-studied microscopic model in evolutionary theory is Kauffmans NK model (*Kauffman and Levin, 1987*), but similar microscopic models can be obtained systematically via Fourier expansion (*Neher and Shraiman, 2011*). This approach leads to the class of spin glass models well-studied in statistical physics (*Sherrington and Kirkpatrick, 1975*; *Sompolinsky and Zippelius, 1982*; *Arous et al., 2001*), theoretical neuroscience (*Amit et al., 1985a*; *Amit et al., 1985b*; *Hopfield, 1982*), ecology (*Roy et al., 2020*), machine learning (*Choromanska et al., 2015*), and combinatorial optimization (*Mézard and Montanari, 2009*; *Mezard et al., 1986*). Recently, spin glass models have been used to study the role of epistasis in adaptive dynamics, leading to insight into the generation of slow, logarithmic fitness trajectories (*Guo et al., 2019*) and into how macroscopic epistasis emerges from widespread microscopic interactions (*Reddy and Desai, 2021*). But due to the computational expense of full-scale microscopic simulations, these prior works make the SSWM assumption, which greatly simplifies the resulting adaptive dynamics to a process that is analytically tractable.

In order to understand the role played by microscopic epistasis and clonal interference in real-world evolving populations, we present a systematic numerical study of the evolutionary dynamics of a microbial culture under serial dilution on a microscopic fitness landscape far from the SSWM regime. To do so, we consider a spin glass type model that enables us to independently tune the magnitude of epistasis and the level of clonal interference in the culture. The model contains both additive and epistatic terms. The relative magnitude of the two terms can be adjusted, and the epistatic interaction takes place pairwise between two loci on the genome.

To resolve the adaptive dynamics, we develop a high-performance, OpenMP-based multi-threaded implementation of the resulting stochastic process in C++ (https://github.com/nmboffi/spin_glass_evodyn copy archived at *Boffi, 2023*), which we use to study an adaptive walk to a fitness peak comprising a few hundred fixation events. The implementation leverages several algorithmic advances to capture the complete microscopic details of the process over long timescales: an efficient algorithm for computation of the fitness that leverages the structure of the epistatic interaction, a hashing-based method for storage of strains by genotype for fast splitting and joining, and an efficient approach for diluting the culture. These algorithmic advances render tractable the computation of the entire distribution of fitness effects, the complete sequence of fixed mutations along with their individual effects, and the number of remaining beneficial mutations for any strain at any time with realistic population sizes (one hundred million bacteria) over realistic timescales (tens of thousands of generations). In the strong clonal interference regime, tracking all microscopic details generates hundreds of gigabytes of data; the resulting datasets are processed by custom-built Python code that produces standard, experimentally-measurable observables.

Our framework brings insight into both real-world experimental systems and modern approaches in evolutionary theory. To this end, we show that hill-climbing dynamics on a random and sparse fitness landscape with two-point interactions cannot give rise to the slow, low-exponent power law trajectories observed in Lenskis LTEE even with clonal interference, suggesting that other factors such as structured interactions might be at play. Moreover, we show that any macroscopic fitness-parameterized model used to describe a microscopic process must depend intrinsically on the level of clonal interference in the population, implying that the DFE in a macroscopic model for an experimental system must be tuned to the mutation rate of the culture.

The paper is organized as follows. In 'Model details', we describe details of the fitness landscape and the simulation. In 'Landscape ruggedness slows the fitness trajectory', we show how microscopic epistasis slows the functional form of the fitness trajectory independent of the level of clonal interference. In 'A fitness-parameterized mapping' and 'A random walk model', we develop simplified macrosopic models to interpret and explain mechanistically the effect of microscopic epistasis. In 'The effect of clonal interference', we show how the effect of clonal interference depends on the strength of microscopic epistasis, and that an accurate fitness parameterized model must be tuned to the level of clonal interference in the population. We conclude with a discussion in 'Discussion and conclusions' *Equation 1*.

## Model details

### Definition

To study the effect of microscopic epistasis on the average fitness trajectory, it is useful to model the genome as a sequence of sites and to consider fitness landscapes that specify the fitness as a function of the state of the genome. We study a generic finite-sites microscopic model (*Figure 1*) inspired by the Sherrington Kirkpatrick spin glass in statistical physics (*Sherrington and Kirkpatrick, 1975*). In the next subsection, we elaborate on the generality of a model of this form.

### The fitness model

With $L$ denoting the length of the genome, the fitness of a strain with genotype $\boldsymbol{\alpha} \in \{\pm 1\}^L$ is given by the expression

$$F(\boldsymbol{\alpha}) = \sum_{i=1}^{L} h_i \alpha_i + \sum_{i<j}^{L} \alpha_i J_{ij} \alpha_j + F_{off}. \tag{1}$$

Here, $F_{\text{off}}$ is an arbitrary offset value that can be used to fix the initial fitness independent of the initial genotype. Inspired by experimental competition assays in typical laboratory microbial evolution settings, we compare the fitness of a given strain to the fitness of the ancestral strain. To do so, we choose $F_{\text{off}}$ so that the fitness of the ancestral strain is equal to one; this arbitrary shift has no effect on the dynamics or our conclusions.

In *Equation 1*, each $h_i$ represents the instantaneous contribution of a mutation at gene $i$ to the fitness of the strain in the absence of epistasis. Each $J_{ij} = J_{ji}$ describes the microscopic epistasis between mutations at genes $i$ and $j$. Realistic biological fitness landscapes are thought to be rugged,

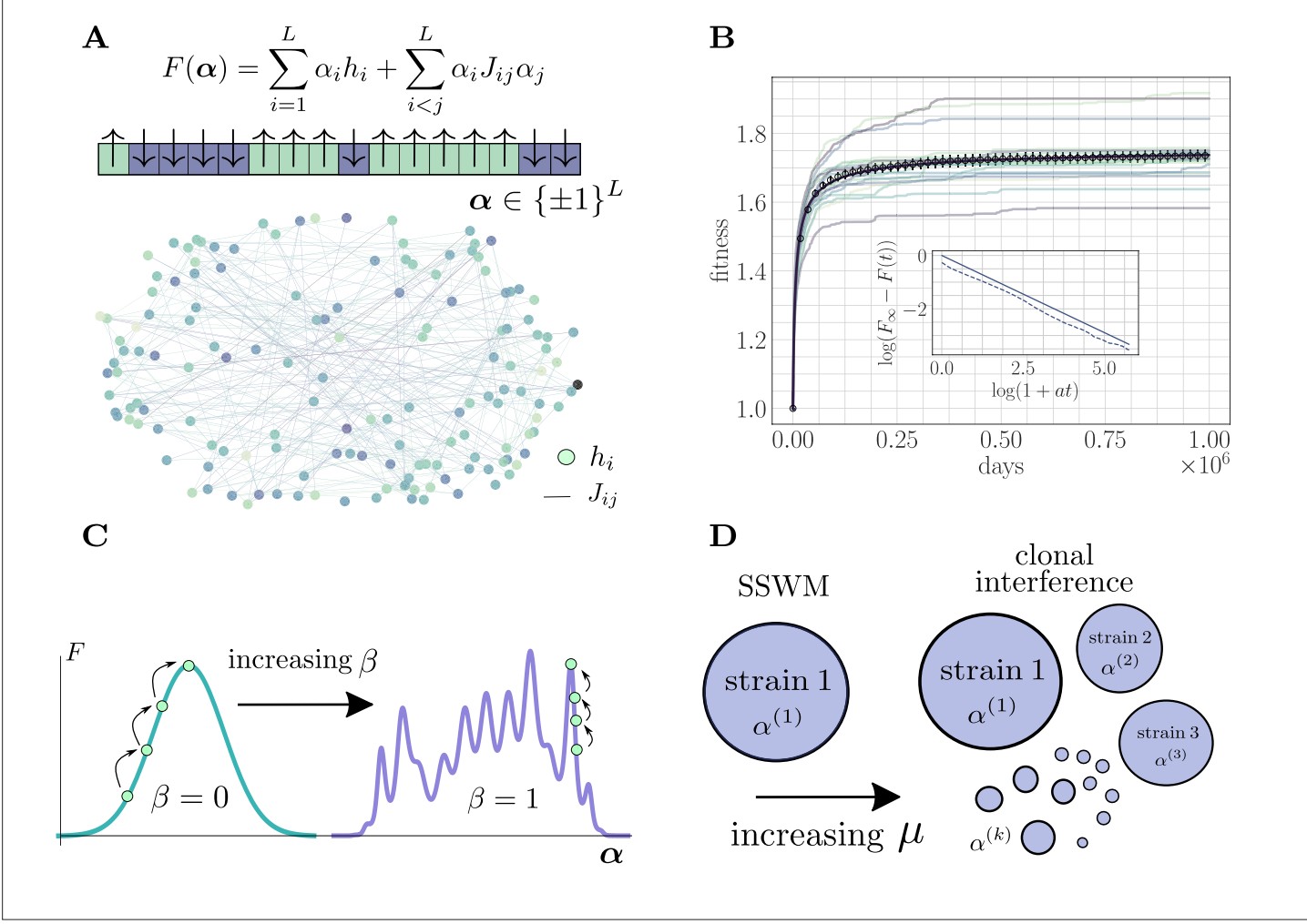

**Figure 1.** The microscopic epistasis model. (**A**) The fitness of a given strain is defined by its genotype $\boldsymbol{\alpha} \in \{\pm 1\}^L$. Each gene $i$ contributes $\alpha_i h_i$ independent of the background genotype and an epistatic contribution $\alpha_i \sum_{j \neq i} \frac{1}{2} J_{ij} \alpha_j$ due to its interaction with all other genes. The $h_i$ and $J_{ij}$ values are drawn randomly, and the relative magnitudes of the two contributions can be tuned by adjusting a parameter $\beta \in [0, 1]$. Each gene interacts, on average, with a fraction $\rho$ of other genes. The fitness landscape is thus described by a disordered network, here shown for $L = 250$ and $\rho = 0.05$, with color indicating magnitude and connectivity demonstrating the sparsity pattern of the $J_{ij}$. (**B**) A typical hill-climbing trajectory with both clonal interference ($\mu = 2 \times 10^{-4}$) and microscopic epistasis ($\beta = 0.5$). Replicate trajectories are displayed in low opacity, while the mean over all replicates is shown in open circles with error bars depicting the standard error of the mean. The mean fitness is consistent with a power-law relaxation with exponent $c \approx 0.575$ (solid). The inset displays the trajectory (dashed) and the best-fit power law (solid, shifted for visual clarity) on a log-log scale. (**C**) As β is increased, the epistatic contribution becomes more significant, and the landscape smoothly becomes more rugged. (**D**) As the mutation rate increases, the magnitude of clonal interference can be smoothly tuned.

containing many local extrema. By taking the values of $h_i$ and $J_{ij}$ to be random, *Equation 1* gives rise to such a complex fitness landscape.

## Disorder statistics

Because biological networks are typically sparse (*Tong et al., 2004*; *Costanzo et al., 2016*), our model is such that each gene only interacts on average with a fraction $0 < \rho \leq 1$ of the other genes. $\rho$ is typically on the order of a few percent in realistic networks, and we set $\rho$ accordingly. These considerations lead to the choice of distributions

$$h_i \sim \mathsf{N}\left(0, \sigma_h^2\right),$$

$$J_{ij} = \gamma_{ij}\psi_{ij}, \quad \gamma_{ij} \sim \mathrm{N}\left(0, \sigma_J^2\right), \quad \psi_{ij} \sim \mathrm{Ber}(\rho), \quad (i \neq j)$$

with $J_{ii} = 0$ for all $i$. Above, $\mathrm{N}(\mu, \sigma^2)$ denotes a normal distribution with mean $\mu$ and variance $\sigma^2$, and Ber($\rho$) denotes a Bernoulli distribution with parameter $\rho$. The variances are set to $\sigma_h^2 = (1 - \beta) \Delta^2$ and $\sigma_J^2 = \dfrac{\beta \Delta^2}{L\rho}$ where $0 \leq \beta \leq 1$ sets the magnitude of microscopic epistasis and $\Delta > 0$ sets the magnitude of generic fitness increments. This scaling with β ensures that the fitness increase at initialization $\dot{F}(t = 0)$ is approximately independent of β. This choice provides a useful setting to qualitatively compare the speed of fitness trajectories in addition to quantitative measures such as fitting parametric functional forms (*Guo and Amir, 2022*). Typical fitness increments in laboratory experiments are on the order of a few percent of the fitness of the ancestral strain (*Barrick et al., 2009*), and we choose $\Delta$ to match this observation. Our conclusions about the roles of epistasis and clonal interference (and their mechanisms) do not depend on the specific choice.

## Fitness trajectories

The fitness landscape is defined by a sparse and random genetic interaction network (*Figure 1A*). Adaptation dynamics produce fitness trajectories (*Figure 1B*) qualitatively consistent with experimental observations of long-term evolution in microbial populations (*Lenski, 2017*), leading to power-law trajectories with exponents between 0.45 and 3.1. For details on how these trajectories and exponents are produced, see 'Further simulation details'.

## Landscape structure

The parameter $\beta$ can be used to continuously tune the ruggedness of the landscape (*Figure 1C*). For $\beta = 0$, there is no microscopic epistasis, and the evolutionary dynamics corresponds to a hill-climbing event towards the single fitness maximum with value $\sum_i |h_i|$. In the opposite extreme for $\beta = 1$, mutations do not have any effect independent of the genetic background, and the landscape is rife with local fitness maxima. In particular, for $\beta > 0$, the landscape exhibits widespread sign epistasis, which is known to be necessary to generate rugged features thought to be present in realistic experimental fitness landscapes (*Weinreich et al., 2005*). By systematically varying this parameter and observing its effect on the average fitness trajectory throughout the approach to a fitness maximum, we quantify the role of microscopic epistasis in *slowing* the functional form (i.e. reducing the power-law exponent) of the fitness trajectory both with and without clonal interference.

The structure of the landscape near the initialization can also be tuned by adjusting the number of available beneficial mutations, or the *rank R*, of the ancestral strain. To remove this source of variability, we fix the rank of the ancestral strain to be identical across all experiments.

## Dilution and selection

We study a batch culture subject to a standard serial dilution protocol (*Lenski et al., 1991*; *Lin et al., 2020*; *Desai et al., 2007*; *Kryazhimskiy et al., 2012*). The population is allowed to grow until the total number of bacteria in the simulation reaches $N_f = D \times N_0$ where $D$ is the dilution factor. When the total population size reaches $N_f$, we say that a day has been completed. At the end of each day, the population is diluted back to size $N_0$ by sampling from a multivariate hypergeometric distribution. Repeated sampling from this distribution is computationally intensive, and we developed an efficient approximate scheme to do so in the strong clonal interference regime (see 'Further simulation details').

## Simulation parameters

Motivated by Lenski's long-term evolution experiment (*Lenski et al., 1991*; *Lenski and Travisano, 1994*), we set $D = 100, N_0 = 10^6$, and $N_f = 10^8$ in all simulations. We fix $\rho = 0.05$ and $\Delta = 0.005$, respectively motivated by sparsity of biological networks and the overall magnitude of typical fitness increments in laboratory experiments. We set $L = 1000$, which corresponds to considering mutations at the level of each gene in *E. coli*. Because typically only a fraction of possible mutations are beneficial, we fix the initial rank to 100. The mutation rate μ and epistatic parameter β are varied and will be specified along with the results.

## Choice of model

There are several compelling reasons to study the fitness model in *Equation 1* that we now highlight.

### Generality

*Equation 1* represents the two lowest-order terms in a Fourier expansion of an arbitrary function defined on the Boolean hypercube (*Neher and Shraiman, 2011*). As such, our model represents a rigorously quantifiable approximation of *any* choice of fitness model defined on the hypercube.

### Tunable epistasis

The presence of the continuous parameter $\beta$ allows us to systematically vary the relative contribution of the epistatic interaction, enabling a detailed study of the effect of microscopic epistasis on the dynamics of adaptation. The well-known NK model (*Kauffman and Levin, 1987*) similarly contains a parameter ($K$, the number of genes in an epistatic interaction) that can be used to tune the ruggedness of the landscape. However, this parameter is discrete and changing it leads to a more drastic shift in the structure of the fitness landscape.

### Tunable clonal interference

For low mutation rate, the population is in the SSWM regime and the evolutionary dynamics correspond to sequential sweeps of beneficial mutations throughout the population, which consists of a single dominant strain (*Figure 1D*). As the mutation rate increases, the adaptation dynamics become richer, enabling higher-order effects such as multiple mutations (*Weissman et al., 2009*), stochastic tunneling (*Iwasa et al., 2004*; *Guo et al., 2019*), and competing populations, all of which emerge naturally in our simulations.

### Efficiency

*Equation 1* has significant computational advantages–summarized by Lemma 1–that enable us to develop a large-scale simulation. Leveraging algebraic simplifications intrinsic to the model's mathematical structure, the fitness of a given strain may be computed as a correction to the fitness of a reference strain that is updated adaptively to track the state of the population. The resulting adaptive fast fitness computation is several orders of magnitude faster than a naive calculation, and our ability to simulate to long times in the strong clonal interference regime hinges upon it.

## Results

## Landscape ruggedness slows the fitness trajectory

We first study the effect of microscopic epistasis on the functional form of the fitness trajectory in both the SSWM ($\mu = 10^{-8}$) and clonal interference ($\mu = 2 \times 10^{-4}$) mutation regimes (*Figure 2*). Intuitively, by complicating the fitness landscape and increasing the difficulty of the corresponding optimization problem, we expect greater levels of microscopic epistasis to lead to a slower fitness trajectory. Empirically, we find that the value of the fitness peak increases slightly with increasing β. To eliminate this variability, each mean fitness trajectory is normalized to lie between the values 1 and 2 for visualization. The fitness trajectory takes more time to approach its asymptotic value as β increases, indicating a slower approach towards equilibrium (*Figure 2A/B*).

Insight into the mechanism by which epistasis slows the fitness trajectory can be obtained by visualizing the substitution trajectories (*Figure 2C/D*), which describe the number of mutations that have fixed in the population at time $t$. The substitution trajectories demonstrate that increasing the amount of microscopic epistasis smoothly leads to an accumulation of more fixed mutations at each time. Because the initial rank (the number of available beneficial mutations) is identical for each value of β, the substitution trajectories suggest a simple picture: as β increases, a greater number of new available beneficial mutations are generated per each typical fixation event. Moreover, because more mutations are needed to cease adaptation, each typical fixation event must provide less progress towards the fitness peak. Mathematically, this corresponds to a greater prevalence of flat regions in the fitness landscape, which have been identified as a source of slow dynamics in previous studies of spin-glass physics (*Kurchan and Laloux, 1996*).

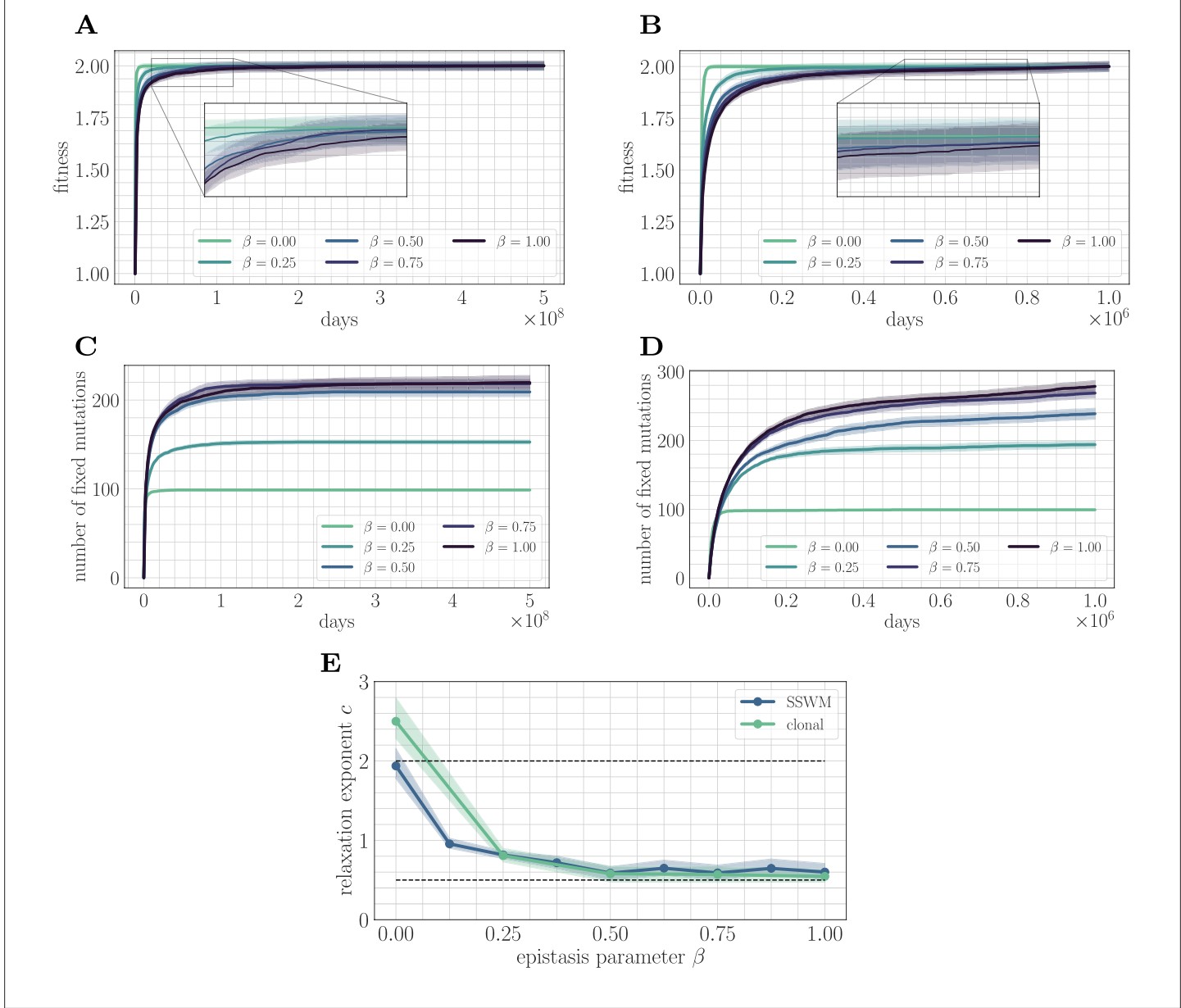

**Figure 2.** Increasing landscape ruggedness slows the fitness trajectory. Left: SSWM. Right: Clonal interference. Error bars for the first four panes (shown as shaded regions, for this and other figures) indicate standard error of the mean over replicates. (**A**) Normalized fitness trajectories as a function of $\beta$ in the SSWM limit ($\mu = 10^{-8}$). Higher values of β exhibit a slower approach towards the fitness peak. (**B**) Normalized fitness trajectories in the clonal interference regime ($\mu = 2 \times 10^{-4}$). (**C**) Substitution trajectories in the SSWM limit ($\mu = 10^{-8}$). Higher values of β lead to a greater number of fixed mutations. (**D**) Substitution trajectories in the clonal interference regime ($\mu = 2 \times 10^{-4}$). (**E**) Fitness relaxation exponents as a function of $\beta$ in the SSWM limit and clonal interference regimes. As $\beta$ increases, the fitness relaxation slows. Dashed lines indicate analytically computable exponents $c = 2.0$ for $\beta = 0$ and $c = 0.5$ for $\beta = 1.0$ in the SSWM regime. Error bars indicate 95% quantiles computed from the bootstrap distribution (for details, see 'Further simulation details').

These observations highlight the role of microscopic epistasis in slowing down long-term evolutionary dynamics, but they do not make a quantitative claim about the functional form of the fitness trajectory. To make such a claim, we can fit the data to a predictive model and study how the model parameters depend on β. The functional form of the fitness trajectory can be computed analytically in the SSWM regime in a fitness-parameterized context approximately met by our fitness model with $\beta = 0$ (***Good and Desai, 2015***); the resulting trajectory is given by the power-law relaxation

$$F(t) = F_\infty - \frac{F_\infty - F_0}{(1 + at)^c} \tag{2}$$

with $F_\infty = \sum_i |h_i|$, $c = 2$, and $F_0 = 1$ by our choice of $F_{\text{off}}$. We find empirically that in both the SSWM and clonal interference regimes, the power-law in *Equation 2* provides a good fit to the mean fitness trajectory for $\beta > 0$.

We fit *Equation 2* to the mean fitness trajectory (see 'Further simulation details') over a range of values of β (*Figure 2E*). In both regimes, the relaxation exponent $c$ decreases monotonically with increasing β, indicating a quantitative slowdown of the fitness trajectory with increasing levels of microscopic epistasis. In the next section, we will show that we can estimate the exponent $c = 0.5$ for $\beta = 1.0$ in the SSWM regime.

## A fitness-parameterized mapping

The mechanism suggested by the substitution trajectories can be confirmed by mapping the microscopic model to a fitness-parameterized landscape with two effective parameters: a single beneficial fitness increment given by the expected beneficial fitness increment $\langle \Delta F_b \rangle$, and a beneficial mutation rate set by the rank $R$ (*Figure 3*). The reduction to a few-parameter fitness-parameterized model has been justified both theoretically (*Good et al., 2012*) and experimentally (*Hegreness et al., 2006*) in the clonal interference regime, and we find that it similarly provides useful insight in the SSWM regime.

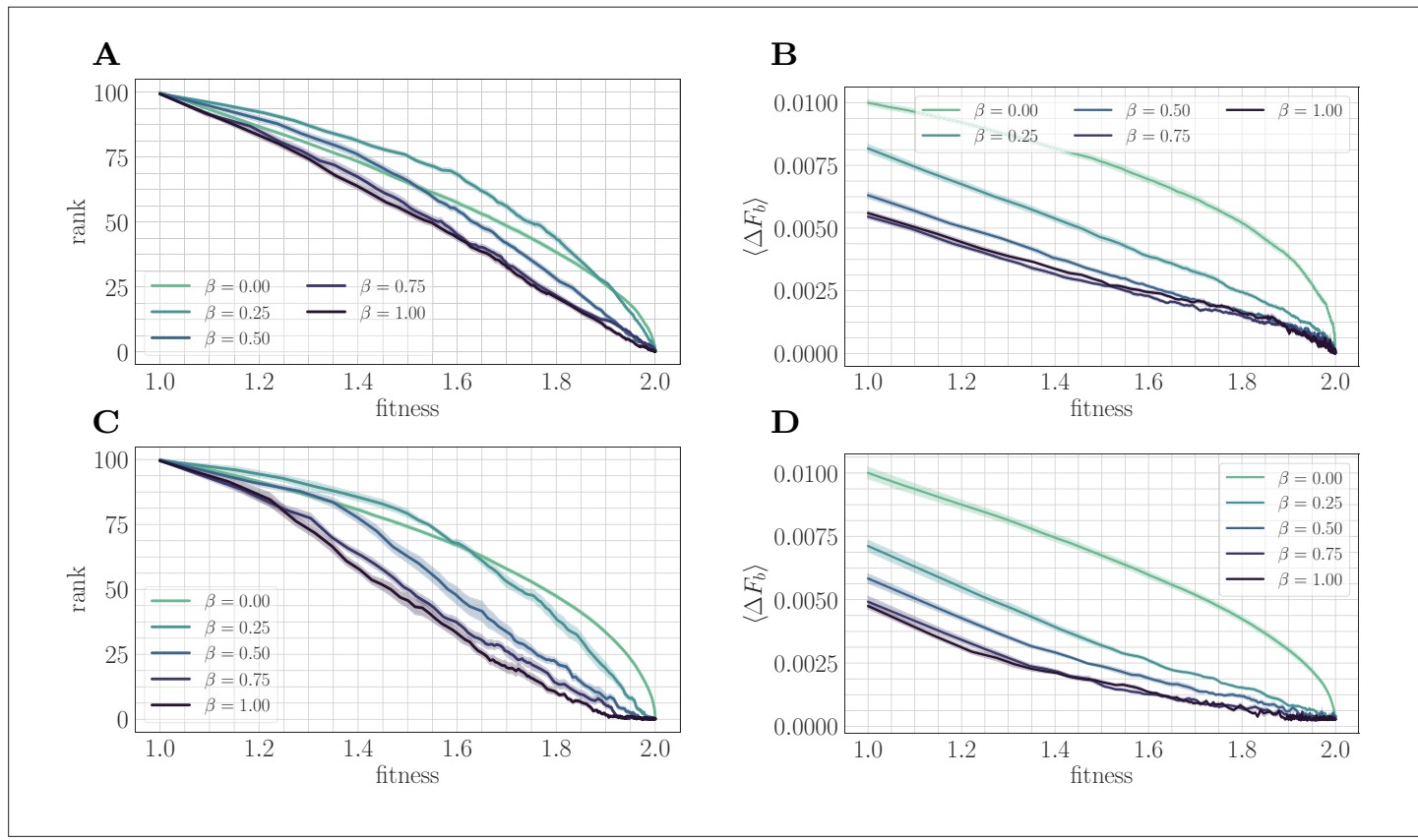

**Figure 3.** A mapping to a fitness-parameterized landscape as a function of $\beta$. Top: SSWM. Bottom: Clonal interference. Error bars in the first four panes indicate standard error of the mean over replicates. (**A**) Rank as a function of fitness in the SSWM regime. The functional form of $R(F)$ changes significantly as β varies, progressing from a concave decreasing function for $\beta = 0.25$ towards a linear form for $\beta = 1$. For all $\beta \neq 0$, the rank decreases monotonically with $\beta$. (**B**) Expected normalized beneficial fitness increment $\langle \Delta F_b \rangle$ as a function of normalized fitness in the SSWM regime. Global epistasis emerges naturally in our model, leading to a linear decrease in $\langle \Delta F_b \rangle$ with $F$ for all nonzero values of β. For all values of β, the increment decreases monotonically with β. (C/D) Rank/expected increment as a function of fitness with clonal interference; similar observations hold as for SSWM.

In both mutation regimes, for all nonzero values of β, the expected beneficial increment behaves linearly as a function of fitness (*Figure 3B/D*). This phenomenon is known as macroscopic or global epistasis, and has been shown to be an emergent property of a class of finite-sites models similar to the one considered here (*Reddy and Desai, 2021*). Consistent with the observations of the previous section, the expected beneficial increment (scaled relative to the fitness peak) decreases with $\beta$ at fixed fitness, which highlights that microscopic epistasis tends to reduce the progress towards the peak provided by each typical fixation event.

Unlike the beneficial fitness increment, the rank $R(F)$ exhibits more variation in functional form as β is varied, which progresses towards linearity as β tends to one (*Figure 3A/C*). Considering only $\beta > 0$, $R(F)$ decreases with increasing β at each $F$; including $\beta = 0$, this is also true for sufficiently large $F$. This indicates that at any fixed relative distance from the fitness peak, beneficial mutation events become more rare as $\beta$ increases.

Taken together, these observations demonstrate that microscopic epistasis leads to a slower fitness trajectory through two complementary effects. At each fixed value of fitness, beneficial mutations are less likely to be found by random mutation; moreover, more of them are required to reach the peak due to a lower typical (normalized) increment. These additional beneficial mutations are generated by the epistatic interaction as each fixation event occurs.

## Quantitative model

These arguments can also be justified mathematically, leading to a quantitative prediction of $c = 0.5$ for $\beta = 1$, consistent with the result in *Figure 2E*. In such a two-parameter macroscopic model, the fitness evolves according to the dynamics

$$\dot{F}(t) \sim \langle \Delta F_b \rangle \langle p_{\text{fix}} \rangle R, \tag{3}$$

where each quantity on the right-hand side is evaluated at $F(t)$. In the SSWM regime, $\langle p_{\text{fix}} \rangle(t) \sim \frac{\langle \Delta F_b \rangle(t)}{F(t)}$ according to Haldane's formula (*Haldane, 1927*). The preceding paragraphs demonstrate that $\dot{F}$ decreases as a function of β for each fixed $F$, giving rise to a slower trajectory. From *Figure 3A/B*, both $R(F)$ and $\langle \Delta F_b \rangle(F)$ are approximately linear and reach zero at $F = 2$. Hence $R \approx k \langle \Delta F_b \rangle$ for a fixed $k > 0$. *Equation 3* then reads

$$\dot{F}(t) \sim \frac{\left(2 - F(t)\right)^3}{F(t)},$$

which predicts that asymptotically

$$F(t) \sim 2 - \frac{B}{(1 + at)^{1/2}},$$

for a fixed $B > 0$. This prediction provides a complement to the analytical result of $c = 2$ in the SSWM regime with $\beta = 0$.

## A random walk model

The previous sections provided an explanation for the effect of microscopic epistasis on long-term adaptation dynamics: as the level of microscopic epistasis increases, the typical number of available beneficial mutations at a given fitness decreases while the number of fixed mutations required to reach the peak increases, leading to a slower trajectory. In this section, we formulate a mechanistic model that provides a qualitative explanation for why microscopic epistasis generates new beneficial mutations and slows the trajectory.

## Distribution of increments

The previous sections highlighted that the mechanism is common to both the SSWM and clonal interference regimes, for which reason we restrict to the SSWM limit in the subsequent analysis. In the SSWM limit, a single empirical distribution of fitness effects is induced by the fitness landscape

$$\rho_t(\Delta F) = \sum_{i=1}^{L} \delta \left( \Delta F - \Delta F_i(t) \right), \tag{4}$$

where $\Delta F_i(t) = -2\alpha_i(t) \left( h_i + \sum_j J_{ij}\alpha_j(t) \right)$ is the fitness effect of a mutation at gene $i$ at time $t$. Because there is only a single strain, the dynamics of adaptation can be characterized entirely by the evolution of $\rho_t$ in time. When a mutation at site $i$ fixes (which can only occur if $\Delta F_i(t) > 0$), the corresponding increment is updated:

$$\Delta F_i \mapsto -\Delta F_i. \tag{5}$$

In the absence of microscopic epistasis, each such fixation event would decrease the rank by one until all available beneficial mutations have fixed. However, due to microscopic epistasis, the fixation of a mutation at gene $i$ causes a change in all other fitness increments:

$$\Delta F_j \mapsto \Delta F_j + 4\alpha_i\alpha_j J_{ij}, \quad j \neq i. \tag{6}$$

The update to the fitness increment in *Equation 6* is complex due to the presence of correlations between $\alpha_i$ and $\alpha_j$ induced by the coupling $J_{ij}$. Moreover, the distribution of $\alpha_i\alpha_j J_{ij}$ must be conditioned on the event that $\Delta F_i > 0$. Previous studies in spin glass (*Horner, 2007*; *Eastham et al., 2006*) and electron glass (*Mogilyanskii and Raikh, 1989*; *Amir et al., 2008*) physics have obtained significant physical insight by neglecting these dynamical correlations, and here we take a similar approach.

### Update statistics

The neglect of dynamical correlations implies that the effect of each fixation event is to add a random Gaussian noise term with probability $\rho$ (the network sparsity parameter) to all other increments,

$$\Delta F_j \mapsto \Delta F_j + \eta_j, j \neq i, \tag{7}$$

$$\eta_j = \gamma_j \psi_j, \quad \gamma_j \sim \mathsf{N}\left( \mu_\beta, \sigma_\beta^2 \right), \quad \psi_j \sim \mathsf{Ber}(\rho).$$

Above, $\mu_\beta$ and $\sigma_\beta^2$ are mean and variance parameters of the noise distribution; these can be estimated numerically from data to account for initialization from a state with $R = 100$ and to condition on beneficial mutation events (see 'Further simulation details'). This process corresponds to a biased random walk (*Equation 7*) with nonlocal transport (*Equation 5*) on the fitness increments.

### Coarse-grained simulation

To test this mechanistic model, we developed a coarse-grained simulation methodology based on Gillespie's stochastic simulation algorithm (*Gillespie, 1976*) (see 'Further simulation details'). Fitting the power law in *Equation 2* to the mean fitness trajectories shows that the random walk model predicts a monotonically decreasing exponent and a monotonically increasing number of fixed mutations as a function of β (*Figure 4A*). This is qualitatively consistent with the results of applying the same coarse-grained simulation approach in the SSWM approximation to the full fitness model (*Figure 4B*). Due to the neglect of correlations, the random walk model predicts a lower number of generated beneficial mutations and a correspondingly higher fitness exponent.

### Why are beneficial mutations generated?

Because only beneficial mutations can fix in the SSWM limit, the transport in *Equation 5* is asymmetric, and beneficial mutations will be rapidly converted to deleterious mutations with equal magnitude but opposite sign. The noise in *Equation 7* broadens the distribution of fitness increments, which converts deleterious mutations with small magnitude into beneficial mutations with small magnitude and vice-versa. The buildup of deleterious mutations results in a diffusive flux from the deleterious half to the beneficial half, providing a simple mechanism for the formation of new beneficial mutations (*Figure 4C*). Empirically, we find that $\sigma_\beta^2$ increases with $\beta$, driving the generation of a greater number of mutations with increasing $\beta$ and a correspondingly slower trajectory.

## The effect of clonal interference

Clonal interference is known to reduce fixation probabilities (*Gerrish and Lenski, 1998*; *Lin et al., 2020*), and hence to slow down the rate of adaptation when compared to an SSWM model with the same parameters. Given its prevalence in realistic evolving laboratory populations, models incorporating both clonal interference and macroscopic epistasis have been developed to predict the slow power law fitness trajectory observed in Lenski's long-term evolution experiment (*Wiser et al., 2013*). Despite this interest, the effect of clonal interference on the shape of the fitness trajectory is still poorly understood.

In fitness-parameterized models, for sufficiently weak clonal interference, the way clonal interference affects the shape of the fitness trajectory depends on how the beneficial mutation rate changes with fitness (*Guo and Amir, 2022*). In particular, for typical models where beneficial mutations become less prevalent as the population climbs up the hill, clonal interference *accelerates* the fitness trajectory. While this result provides insight into the role of clonal interference in laboratory populations, its not clear if it holds more generally in a microscopic framework, or when the magnitude of clonal interference becomes large. Here, we demonstrate that the effect of clonal interference on the fitness trajectory depends on the strength of microscopic epistasis. In its absence, clonal interference is seen

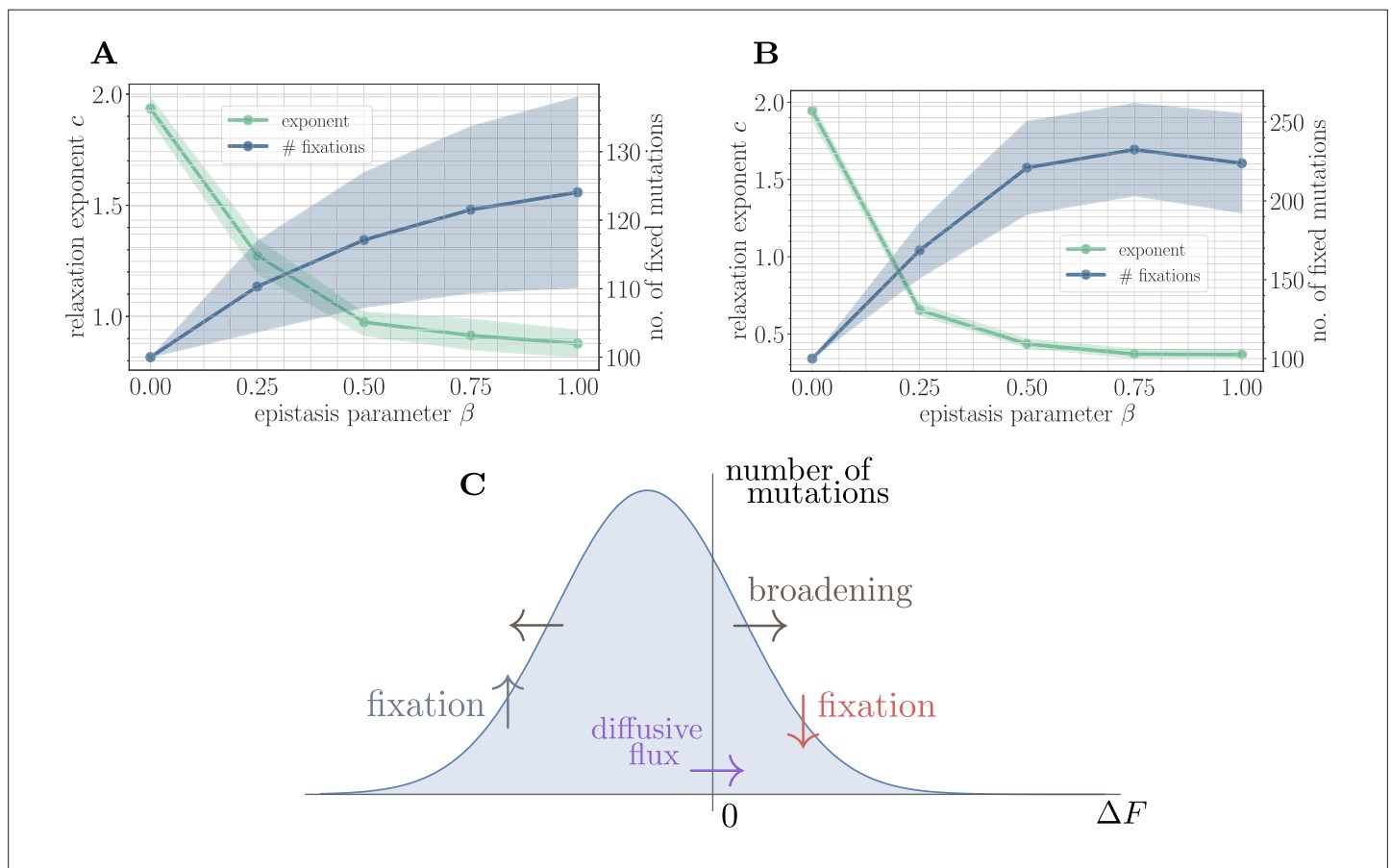

**Figure 4.** A simple random walk model captures the generation of beneficial mutations and a decreasing exponent with increasing microscopic epistasis. (**A**) Fitness relaxation exponent and number of fixed mutations to reach a local fitness maximum (zero rank) as a function of β, computed within a random walk model (which neglects correlations). The model predicts a monotonically decreasing exponent and a monotonically increasing number of fixations as a function of β. Error bars around exponents indicate 95% quantiles from the bootstrap distribution (100 samples) while central line depicts the median. Error bars around the number of fixations indicate the standard deviation over 200 replicates and central line the mean. (**B**) Analogous figure to (**A**), but within the SSWM approximation using a coarse-grained Gillespie simulation framework (where correlations develop over time). The behavior is qualitatively similar to the random walk model, though the exponent decreases further and the number of fixation events is larger. (**C**) Illustration of the diffusive generation of mutations. As beneficial mutations fix, the beneficial half of the distribution of fitness increments $\rho_t(\Delta F)$ is depleted and transported to the deleterious half. Each fixation event causes the distribution to broaden due to the epistatic interaction, which combines with a buildup of deleterious mutations to create a diffusive flux from the deleterious half to the beneficial half of the distribution.

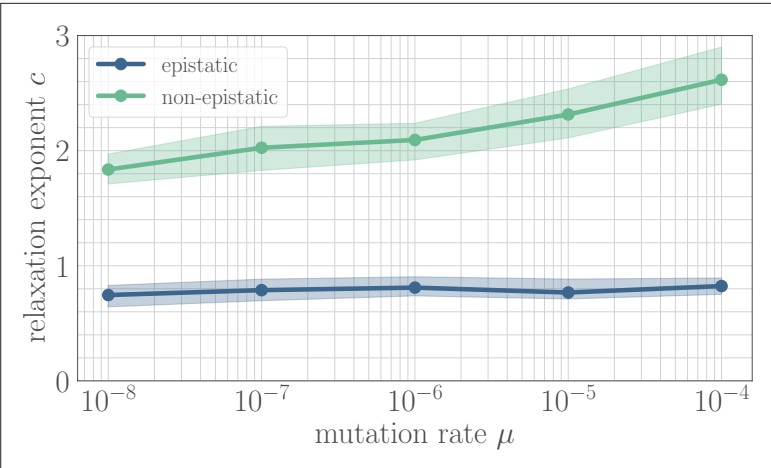

**Figure 5.** The effect of clonal interference on the fitness trajectory depends on the strength of microscopic epistasis. Results of fitting the power law in *Equation 2* as a function of μ. The relaxation exponent $c$ increases monotonically with μ for the non-epistatic model ($\beta = 0$), while the exponent remains essentially constant in the epistatic model ($\beta = 0.25$).

to accelerate the trajectory, while for a sufficiently strong epistatic interaction, clonal interference does not quantitatively affect the speed of the trajectory.

In simulation, the magnitude of clonal interference is tuned by adjusting the mutation rate μ. Increasing the mutation rate increases clonal interference, but also accelerates adaptation by allowing for more mutations each day. This effect can be eliminated by normalizing time by the mutation rate, so as to isolate the role of clonal interference itself. Viewed in these units, clonal interference decelerates the fitness trajectory due to the suppression of fixation probabilities (*Appendix 1—figure 1*). However, when measuring the speed of a fitness trajectory, it is more rigorous to assign a quantitative measure by fitting a functional form such as the power law in *Equation 2*, which will directly estimate the timescale parameter $a$ independently for each μ. Fitting this power law reveals exponents that hover around $c \approx 0.8$ in the epistatic setting (here, $\beta = 0.25$), but increase from $c \approx 1.85$ for $\mu = 10^{-8}$ to $c \approx 2.6$ for $\mu = 10^{-4}$ in the non-epistatic setting (*Figure 5*). These results can also be visualized qualitatively by normalizing time in units such that the initial rate of fitness increase is constant for different values of μ, as was done for the results in *Figure 2* via the definition of $\beta$ (*Appendix 1—figure 2*).

## A change in the effective landscape

To understand this result and why it differs from the predictions of the fitness-parameterized setting, we can map the microscopic model to an effective fitness-parameterized landscape (*Figure 6*). This mapping reveals a striking observation: the effective macroscopic model depends on the mutation rate, which is typically treated as an independent parameter in the standard fitness-parameterized framework. Intuitively, there are many states in the landscape with the same value of $F$ but which differ in their conditional distribution of increments $\rho(\Delta F|F)$; these states are dynamically selected in a way that depends on the level of clonal interference. In both the epistatic and non-epistatic landscapes, we find that the distribution becomes more sharply peaked for higher levels of clonal interference, while it has a heavier tail for lower levels (*Figure 6A/B*). This occurs because clonal interference suppresses the fixation of low-effect mutations. The result is that systems with high clonal interference typically reach a given $F$ through fewer, more valuable mutations, while systems with low clonal interference typically reach $F$ through the accumulation of more low-value mutations.

This effect can be visualized over the trajectory by making use of the summary statistics $\langle \Delta F_b \rangle$ (expected increment) and $R$ (rank) viewed as a function of the fitness. In both landscapes, the expected increment decreases with increasing μ (*Figure 6C/D*). By contrast, the behavior of the rank depends on the strength of microscopic epistasis. Without epistasis, the rank increases with μ (*Figure 6E*), while with epistasis, the rank becomes non-monotonic in μ, and becomes comparable at high fitness (*Figure 6F*). This observation highlights the importance of the rank in setting the speed of the fitness

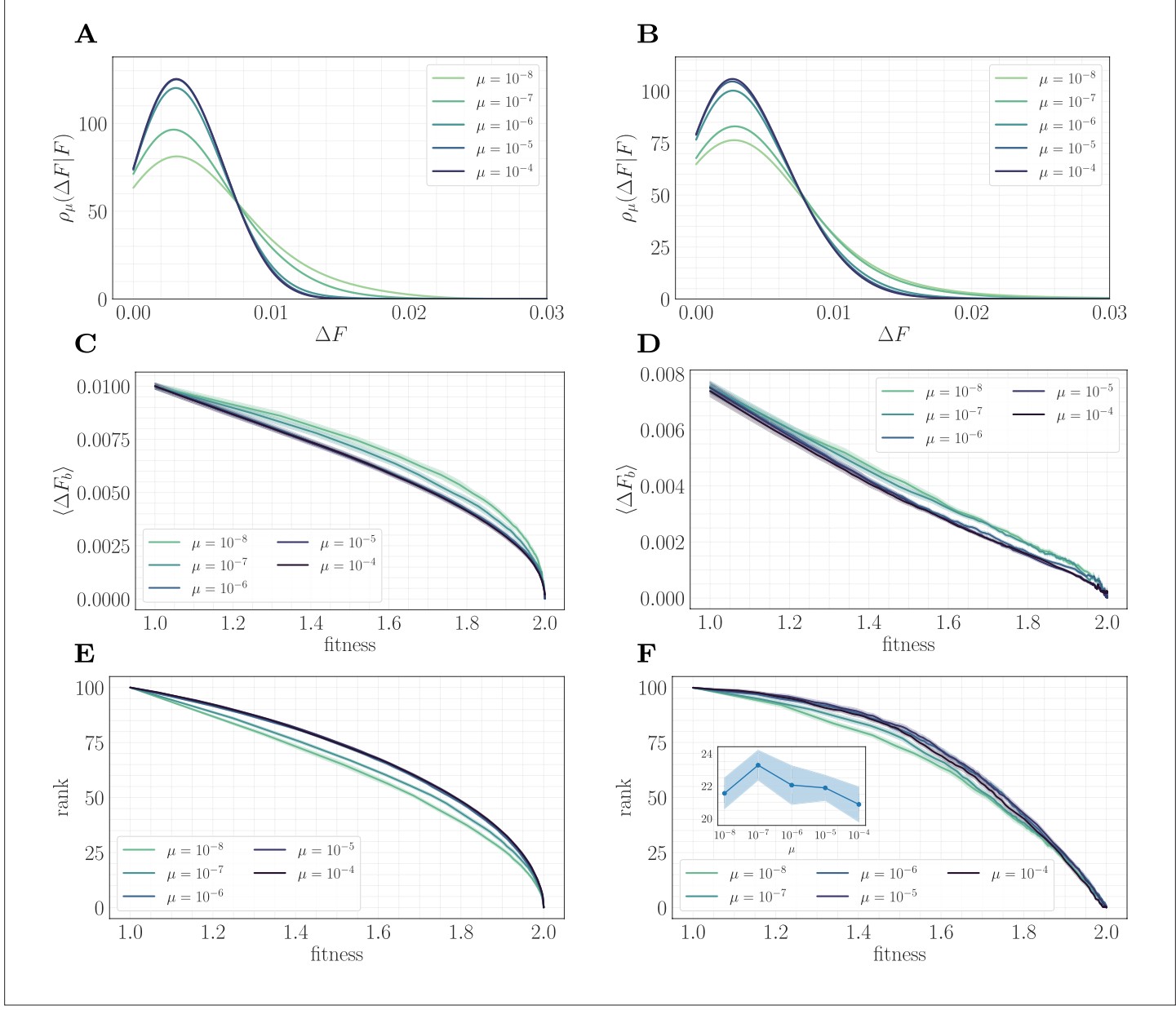

**Figure 6.** A mapping to a fitness-parameterized landscape as a function of µ. Left: non-epistatic ($\beta = 0$). Right: epistatic ($\beta = 0.25$). (A/B) Unlike in classical fitness-parameterized models, the level of clonal interference cannot be tuned independently from the landscape. Because clonal interference modifies fixation probabilities, the conditional distribution of fitness effects $\rho_\mu(\Delta F|F)$ depends on µ both with and without epistasis (epistatic: $F \approx 1.85$; non-epistatic: $F \approx 1.65$. Distributions were approximated via kernel density estimation with a bandwidth parameter $\sigma = 1.0$). (C/D) Expected beneficial increment as a function of fitness. Both with and without epistasis, the expected increment tends to decrease with increasing µ at fixed fitness, capturing the fact that higher µ favors the fixation of mutations with higher effect. (E/F) Average rank as a function of fitness. Without epistasis, the rank increases with increasing µ at fixed fitness. With epistasis, the rank curves behave non-monotonically in µ at fixed fitness, and become comparable at high fitness. Inset in (F) shows rank as a function of µ for fixed $F \approx 1.9$, highlighting the non-monotonic behavior.

trajectory, analogous to the results presented in 'A fitness-parameterized mapping'. Both with and without microscopic epistasis, the behavior of the rank with $\mu$ parallels that of the exponent $c$.

## Discussion and conclusions

In this paper, motivated by laboratory serial dilution experiments, we developed a high-performance simulation approach to study the dynamics of long-term adaptation. We focused on a generic

microscopic model that considers the microbial genome as a collection of sites with a binary value indicating the presence of a mutation. Our model contains a non-dimensional parameter $0 \leq \beta \leq 1$ that enables us to smoothly tune the relative contribution of microscopic epistasis to the fitness effect $\Delta F_i$ of a mutation at gene $i$. In addition, we can tune an overall mutation rate μ to adjust the magnitude of clonal interference in the culture. By independently varying the parameters β and μ, we mapped out a phase diagram that describes the effects of microscopic epistasis and clonal interference on the functional form of the mean fitness trajectory.

Our simulation approach gives us the ability to probe microscopic details that are challenging to obtain experimentally, such as statistics of the distribution of fitness effects and the rank over time. The approach also allows us to study regimes such as strong microscopic epistasis and strong clonal interference that have eluded previous theoretical study. In addition to its generality, our model has computational advantages that enable us to probe the long-time dynamics required to reach a local fitness maximum.

## The role of microscopic epistasis

By mapping the model to a simplified fitness-parameterized landscape, we showed that as the strength of microscopic epistasis increases, more mutations are needed to reach a local fitness maximum. In addition, beneficial mutations become less likely; these two properties together lead to a slower trajectory. We isolated a mechanism for this phenomenon -- the generation of new, low-effect beneficial mutations mediated by the epistatic interaction when the culture is at high fitness -- and we showed through a random walk model that this generation is sufficient to slow the fitness trajectory.

A by-product of our analysis is an observation that, as microscopic epistasis increases in strength, the beneficial mutation rate becomes an increasingly linear function of fitness, similar to the phenomenon of global epistasis observed for the expected beneficial fitness increment (*Reddy and Desai, 2021*). This suggests that the strength of epistasis in realistic microbial populations could be inferred by measuring the beneficial mutation rate as a function of fitness experimentally.

## The role of clonal interference

Through a similar analysis, we observed that in the microscopic context considered here, any equivalent fitness-parameterized model must depend on the mutation rate μ, a parameter that is typically tuned independently. In effect, the change in fixation probabilities induced by clonal interference filters the accessible genotypes with a given fitness $F$, giving rise to a conditional distribution of fitness increments $\rho_\mu(\Delta F|F)$ that depends on μ.

Based on this observation, we showed that the effect of clonal interference on the fitness trajectory differs from prior predictions made in the fitness-parameterized setting, and moreover that it depends on the strength of microscopic epistasis. In a non-epistatic model, increasing clonal interference accelerates the fitness trajectory. For sufficiently strong microscopic epistasis, clonal interference has no effect on the speed of the fitness trajectory. We leave the development of a mechanistic model capturing this phenomenon to future work.

## The beneficial mutation rate

A surprising observation is that the trend of the rank with β in *Figure 3A/B* and with μ in *Figure 6E/F* is consistent with the behavior of the exponent with $\beta$ and $\mu$. This is similar to predictions made within the fitness-parameterized framework, where it was found that the effect of clonal interference depends on how the beneficial mutation rate changes with fitness (*Guo and Amir, 2022*). Taken together, these results suggest that the behavior of the beneficial mutation rate as a function of fitness plays a central role in setting the speed of the fitness trajectory.

## Connections to spin glass physics

In this work, we studied a model inspired by the Sherrington-Kirkpatrick spin glass using the techniques of microbial population genetics. Nevertheless, the fundamental questions we study here -- such as characterizing the speed and functional form of relaxation processes -- are also studied in the spin glass literature using seemingly different tools. In particular, the two- and four-point correlation functions

$$\chi_2(t_w, \Delta t) = \frac{1}{L} \sum_{l=1}^{L} \langle \alpha_i(t_w) \alpha_i(t_w + \Delta t) \rangle$$

and

$$\chi_4(t_w, \Delta t) = \frac{1}{L^2} \sum_{i,j=1}^{L} \langle \alpha_i(t_w) \alpha_i(t_w + \Delta t) \alpha_j(t_w) \alpha_j(t_w + \Delta t) \rangle$$

are often studied as a function of the waiting time $t_w$ and the lag time $\Delta t$ to characterize the decay of spin correlations and the importance of correlated spin flips, respectively (**Castellani and Cavagna, 2005**; **Toninelli et al., 2005**) (Here, angular brackets denote an average over independent trajectories).

It is a simple calculation to show that the two-point correlation function obeys the identity

$$\chi_2(0, \Delta t) = 1 - \frac{2m(\Delta t)}{L} \tag{8}$$

where $m(t)$ denotes the population mean substitution trajectory at time $\Delta t$. For general $t_w$, $\chi_2(t_w, \Delta t)$ simply shifts the definition of the ancestral strain. This relation provides a novel link between standard techniques in spin glass physics and microscopic population genetics. Yet, the dynamics of fixation induce important differences: it is well-known that in the absence of epistasis the substitution trajectory follows a power law relaxation similar to **Equation 2** with $c = 1.0$ (**Good and Desai, 2015**). This stands in contrast to the standard setting in spin glasses, where the two-point correlation function often exhibits stretched exponential relaxations (**Phillips, 1996**).

We fit power law relaxations to the $\chi_2(0, \Delta t)$ trajectories in the SSWM regime as a function of β and found a median exponent 1.03 for $\beta = 0$. For $\beta > 0$, we still found good agreement with a power law functional form and obtained a decaying exponent with increasing β (**Appendix 1—figure 3**). We also found that the average over sites in the definitions of $\chi_2$ and $\chi_4$ are roughly equivalent to the angular average over trajectories, so that $\chi_4(0, \Delta t) \approx \chi_2(0, \Delta t)^2$ (**Appendix 1—figure 4**). This demonstrates that it is sufficient to consider the two-point correlation function, or equivalently the substitution trajectory, and that no additional information is contained in the four-point correlator.

## Future directions

Our model and simulation can be extended in many exciting directions. One possibility is to allow for horizontal gene transfer between bacterial strains. By allowing large jumps across the fitness landscape, horizontal gene transfer may have a similar effect to microscopic epistasis, and could slow the fitness trajectory by generating groups of new beneficial mutations (**Slomka et al., 2020**); on the other hand, it could also accelerate the fitness trajectory by allowing for larger steps towards a fitness maximum. Another possibility is to bias the model, so that the $h_i$ and $J_{ij}$ have non-zero means, and to study the effect of these mean values on the long-term dynamics. A third possibility is to allow for further structure in the interaction matrix $J$, rather than the i.i.d. random entries considered here. For example, by allowing for low-rank structure in $J$, one could in principle quantify the role of connected modules of mutations on the functional form of the fitness trajectory (**Parter et al., 2008**; **Landau et al., 2016**; **Landau and Sompolinsky, 2021**). A final direction would be to form a mechanistic model for the effect of clonal interference both with and without epistasis, similar to the random walk model developed to understand the effect of microscopic epistasis.

Our work focuses primarily on clonal interference between potential mutations, which is the most well-studied form of clonal interference. However, recent work has shown that an alternative within-path clonal interference between a mutant and its ancestor has a significant effect on both the rate of adaptation and on the specific adaptive trajectories selected in evolving populations (**Ogbunugafor and Eppstein, 2016**); within-path clonal interference can be quantified along a given trajectory as the sum of the inverse fitness increments. In the clonal interference regime, **Figure 3D** shows that the expected fitness increment decreases at fixed fitness with increasing β. This implies that at a given fitness, the total level of within-path clonal interference increases with β on average. It would be interesting to study how much this contributes to the decrease in exponent $c$ with increasing β in the clonal interference regime. This could be quantified, for example, by computing the total

within-path clonal interference along potential trajectories and identifying how well it correlates with the preferred trajectories as a function of β.

*Figure 5* shows that for $\beta = 0.25$, clonal interference has no effect on the fitness trajectory. Moreover, *Figure 2E* highlights that this remains true for $\beta > 0.25$. It would be interesting to carefully probe the value of β for which the effect of clonal interference vanishes, and to determine if this occurs as a sharp phase transition or if there is a gradual decay of the effect of clonal interference with β. Understanding this behavior could lead to a way to measure the strength of epistasis in experimental systems, by studying how the fitness trajectory depends on the level of clonal interference.

In addition to the extensions considered above, our simulation environment forms a fertile testing ground for theoretical predictions. The relaxation exponents as a function of β considered in *Figure 2E* could in principle be predicted within the random walk approximation using the techniques developed by *Horner, 2007*, or within a dynamical mean-field theory that studies the evolution of the average value of the mutation variables $\langle \alpha_i(t) \rangle \in [-1, 1]$ over time (*Ginzburg and Sompolinsky, 1994*; *Sompolinsky and Zippelius, 1981*; *Sommers, 1987*). Solving such a model, in addition to analytically quantifying the slowing of the fitness trajectory with β, would provide a prediction for a functional form that could be tested against experimental fitness trajectories.

## Acknowledgements

We thank Haim Sompolinsky, Guy Bunin, and Yoav Ram for many useful discussions, and we thank Jimmy Almgren-Bell for contributions to an initial version of the simulation software. M B was partially supported by the Research Training Group in Modeling and Simulation funded by the National Science Foundation via grant RTG/DMS 1646339. H R was partially supported by the Applied Mathematics Program of the U.S. DOE Office of Science Advanced Scientific Computing Research under Contract No. DE-AC02-05CH11231.

## Additional information

### Competing interests

Ariel Amir: Reviewing editor, *eLife*. The other authors declare that no competing interests exist.

### Funding

| Funder | Grant reference number | Author |
| --- | --- | --- |
| National Science Foundation | RTG/DMS - 1646339 | Nicholas M Boffi |
| U.S. Department of Energy | DE-AC02-05CH11231 | Chris H Rycroft |

The funders had no role in study design, data collection and interpretation, or the decision to submit the work for publication.

### Author contributions

Nicholas M Boffi, Conceptualization, Data curation, Software, Formal analysis, Validation, Investigation, Visualization, Methodology, Writing – original draft, Writing – review and editing; Yipei Guo, Conceptualization, Validation, Methodology, Writing – review and editing; Chris H Rycroft, Conceptualization, Software, Supervision, Validation, Methodology, Writing – review and editing; Ariel Amir, Conceptualization, Formal analysis, Supervision, Funding acquisition, Validation, Methodology, Project administration, Writing – review and editing

### Author ORCIDs

Nicholas M Boffi ![ORCID] https://orcid.org/0000-0003-1336-7568
Ariel Amir ![ORCID] http://orcid.org/0000-0003-2611-0139

Reviewer #1 (Public Review): https://doi.org/10.7554/eLife.87895.3.sa1
Reviewer #2 (Public Review): https://doi.org/10.7554/eLife.87895.3.sa2

Author Response https://doi.org/10.7554/eLife.87895.3.sa3

## Additional files

### Supplementary files
• MDAR checklist

### Data availability
The current manuscript is a computational study, so no data have been generated for this manuscript. Modelling code is available at https://github.com/nmboffi/spin_glass_evodyn/tree/main (copy archived at *Boffi, 2023*).

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

## Appendix 1

### Further simulation details

Here, we provide an overview of the simulation methodology, as well as values for the parameters used in our numerical experiments.

### Strains

We begin with a population of size $N_0$ and with an initial ancestral genotype $\alpha^{(0)}$ drawn randomly as $L$ i.i.d. random variables taking the values ±1 with equal probability. We define mutations with respect to this initial sequence and define a strain by the path taken on the hypercube from the initial genotype to its current genotype. Due to microscopic epistasis, the fitness effect of a mutation can only be defined in the context of its genetic background. Defining strains by their path ensures that mutations at the same site with different fitness effects are correctly tracked throughout the simulation.

### Fitness and growth

We assume that each strain grows exponentially with rate given by its fitness. The primary quantity we study is the population mean fitness $F(t)$, which is defined as the average fitness over all cells in the population. To obtain $F(t)$, we average its value over replicate trajectories initialized from different ancestral genotypes within the same fitness landscape.

Recall that the fitness of a strain with genotype $\alpha \in \{\pm 1\}^L$ is given by

$$F(\boldsymbol{\alpha}) = \sum_{i=1}^{L} h_i \alpha_i + \sum_{i<j}^{L} \alpha_i J_{ij} \alpha_j + F_{off}. \tag{9}$$

Because each strain grows with rate proportional to its fitness, we may write

$$N_i(t + \Delta t) = N_i(t) e^{F(\boldsymbol{\alpha}^{(i)})\Delta t}. \tag{10}$$

In *Equation 10*, $N_i(t)$ represents the size of strain $i$ at time $t$, $\boldsymbol{\alpha}^{(i)}$ is the genotype for strain $i$, and $\Delta t > 0$ is a timestep (set to $\Delta t = 0.01$ in all simulations). To avoid biasing the growth for strains with low bacteria count, we allow $N_i(t) \in \mathbb{R}$ throughout the simulation.

### Mutations

As the strains grow, mutants are generated with a fixed mutation rate $\mu > 0$, which describes the probability of a cell gaining a mutation when it divides. The genotype of a mutant is obtained from the genotype of the parent by flipping a site uniformly at random. A mutation is said to have *fixed* if it is present in all strains. For a mutant with genotype $\boldsymbol{\alpha}^{(c)}$ produced from a parent with genotype $\boldsymbol{\alpha}^{(p)}$, the fitness increment is $\Delta F\left(\boldsymbol{\alpha}^{(c)}, \boldsymbol{\alpha}^{(p)}\right) = F\left(\boldsymbol{\alpha}^{(c)}\right) - F\left(\boldsymbol{\alpha}^{(p)}\right)$, while the selection coefficient is $s(\boldsymbol{\alpha}^{(c)}, \boldsymbol{\alpha}^{(p)}) = \Delta F(\boldsymbol{\alpha}^{(c)}, \boldsymbol{\alpha}^{(p)})/F(\boldsymbol{\alpha}^{(p)})$.

After a step of size $\Delta t$, following *Equation 10*, the number of new bacteria produced by a given strain is equal to

$$\Delta N_i(t) = N_i(t) \left( e^{F(\boldsymbol{\alpha}^{(i)})\Delta t} - 1 \right).$$

To ensure that the number of mutants generated by strain $i$ does not exceed the number of bacteria $\Delta N_i(t)$ generated by strain $i$, we draw a Poisson random variable

$$K_i(t) = \text{Poiss}\left(\Delta N_i(t)\mu\right),$$

and then set the number of mutants to be

$$M_i(t) = \min\left(K_i(t), \Delta N_i(t)\right).$$

As new strains are generated through mutation events, we must check if they already exist in the population. If the strain already exists, the mutant joins the existing strain rather than defining a new one. Checking for the existence of a newly generated strain in the overall population can be

performed efficiently by hashing the list of integers defining the path through genome space. All current paths can be stored in a set defining the active strains: the time complexity of checking set membership scales as $\mathcal{O}(1)$ with the number of strains, which is a significant reduction compared to checking all new mutant strains against all existing strains.

## Dilution protocol

The number of bacteria in each strain that make it through dilution to the following day follows a multivariate hypergeometric distribution. To efficiently sample from this distribution, we sequentially draw from the hypergeometric marginal distributions (**Gentle, 1998**). We first sort the population by number of bacteria in descending order. We then sample $k_1 \sim \text{Hyper}\left(N_f, N_1, N_0\right)$ where $N_1$ is the size of the largest strain. We then recursively apply this procedure, choosing

$$k_j \sim \text{Hyper}\left(N_f - \sum_{l<j} k_l, N_j, N_0 - \sum_{l<j} k_l\right)$$ until we have drawn $N_0$ bacteria or we have gone through

the entire population. For greater efficiency, each hypergeometric marginal distribution can be replaced with a draw from a binomial distribution. We verified that our results are independent to this approximation.

## Replicates

Each simulation is performed with a number of replicate populations. Each replicate is instantiated with the same quenched disorder as specified by the $h_i$ and $J_{ij}$. The dynamics for each replicate differ through the random initialization of the ancestral genotype $\boldsymbol{\alpha}^{(0)}$ and through the sequence of random mutations. In the experiments studying the role of clonal interference, the same $h$ and $J$ are used as $\mu$ is varied.

## Power-law fitting methodology

To obtain fitness exponents, the mean (computed over replicates) trajectory is fit via nonlinear least-squares using the scipy function curve_fit. The standard error of the mean is used to weight the residuals in the loss function. Error bars are computed via the bootstrap method, by randomly sampling subsets of trajectories and fitting models to the mean over each subset. The corresponding estimates define an empirical distribution over parameters, from which we compute quantiles and use the median as the estimate of the parameters.

## Random walk statistics

We perform the following procedure to obtain an estimate of $\mu_\beta$ and $\sigma_\beta^2$. We average over initial landscapes and n_inits initializations with rank $R$ for each value of β. For each initialization, we select a single beneficial mutation at random and compute the changes to all other $R-1$ beneficial fitness increments. We compute the empirical mean and variance of these changes and average the resulting estimates over all landscapes and initializations. In the numerical experiments reported here, we set $R = 100$, n_inits = 10, and n_landscapes = 10, though we found that our results were insensitive to the choice of number of landscapes and initializations.

## Coarse-grained simulation methodology

To simulate the discrete random walk model, we compute an initial empirical distribution of fitness increments according to **Equation 9** and save the sparsity pattern defined by the randomly drawn $J_{ij}$. At each fixation event, we flip $\Delta F_i \mapsto -\Delta F_i$ and adjust each $\Delta F_j \mapsto \Delta F_j + \eta_j$ for $j \neq i$ as discussed in the main text. If $J_{ij} = 0$ in the originally drawn genetic network, we set $\eta_j = 0$. For comparison, the full SSWM dynamics can be simulated using **Equation 9** directly.

Fixation probabilities are determined by Haldane's formula $p_{\text{fix}}(\Delta F) \sim \Delta F / F$. To fit our discrete random walk model into the framework of Gillespie's stochastic simulation algorithm, we define chemical reactions corresponding to each possible fixation event. The reaction propensity for a mutation at site $i$ is taken to be equal to $p_{\text{fix}}(\Delta F_i)$. We choose a mutation site at random with probability proportional to its fixation probability. We randomly draw the time that occurred before the next fixation event according to an exponential distribution with mean $1/\sum_{i=1}^{L} p_{\text{fix}}(\Delta F_i)$ (up to an overall mutation rate fixing the units of time).

## Fast fitness computation

The following lemma gives a fast algorithm for computing the fitness of a given bacterial strain. The proof proceeds by noting that rather than computing *Equation 1* directly, we can compute the fitness of a strain $\boldsymbol{\alpha}^{(c)}$ with respect to its parent strain $\boldsymbol{\alpha}^{(p)}$. It concludes by observing that the fitness of $\boldsymbol{\alpha}^{(p)}$ can be related to the fitness of an arbitrary reference strain $\boldsymbol{\alpha}^{(r)}$. This reference strain can be adjusted on-the-fly to track the state of the population.

Lemma 1. Let $\boldsymbol{\alpha}^{(r)} \in \{\pm 1\}^L$ denote the genotype of a reference strain, $\boldsymbol{\alpha}^{(p)} \in \{\pm 1\}^L$ denote the genotype of the parent strain, and $\boldsymbol{\alpha}^{(c)} \in \{\pm 1\}^L$ denote the genotype of the child strain obtained from $\boldsymbol{\alpha}^{(p)}$ via a mutation at gene $k$. Then,

$$F(\boldsymbol{\alpha}^{(c)}) = F(\boldsymbol{\alpha}^{(p)}) - 2\alpha_k^{(p)} \left( h_k + \left( \mathbf{J}\boldsymbol{\alpha}^{(r)} \right)_k - 2 \sum_{j \in M_r} J_{kj}\alpha_j^{(r)} \right),$$  (11)

where $M_r$ denotes the set of mutations of the parent with respect to the reference strain.

*Equation 11* states that if we store the elements of the vector $\mathbf{J}\boldsymbol{\alpha}^{(r)} \in \mathbb{R}^L$, we can compute the fitness of a child strain in terms of the fitness of the parent strain in time complexity $\mathcal{O}(|M_r|)$. This is a massive improvement over the $\mathcal{O}(L^2)$ complexity corresponding to a naive calculation of the quadratic form, and a large improvement over the $\mathcal{O}(L)$ complexity corresponding to computing the fitness of the child with respect to the fitness of the parent, particularly if the reference strain is updated to keep $|M_r|$ small. We found empirically that setting the reference strain to be equal to the dominant strain every time the dominant strain accumulates 15 new fixed mutations was a good heuristic.

*Proof.* Observe that we may write

$$F(\boldsymbol{\alpha}^{(c)}) = F(\boldsymbol{\alpha}^{(p)}) - 2\alpha_k^{(p)} \left( h_k + \sum_j J_{kj}\alpha_j^{(p)} \right).$$

Now, write that $\alpha_j^{(p)} = \alpha_j^{(r)} + \delta\alpha_j$ with

$$\delta\alpha_j = \begin{cases} -2\alpha_j^{(r)} & j \in M_r, \\ 0 & \text{else.} \end{cases}$$  (12)

Plugging this in completes the proof.

## Additional figures

*Appendix 1—figures 1 and 2* visualize the fitness trajectory as a function of normalized time, as referenced in the main text.

*Appendix 1—figures 3 and 4* consider the correlation functions

$$\chi_2(\Delta t) = \left\langle \frac{1}{L} \sum_{i=1}^{L} \alpha_i(0)\alpha_i(\Delta t) \right\rangle$$

and

$$\chi_4(\Delta t) = \left\langle \frac{1}{L^2} \sum_{i,j=1}^{L} \alpha_i(0)\alpha_i(\Delta t)\alpha_j(0)\alpha_j(\Delta t) \right\rangle$$

from spin glass physics, as referenced in the main text.

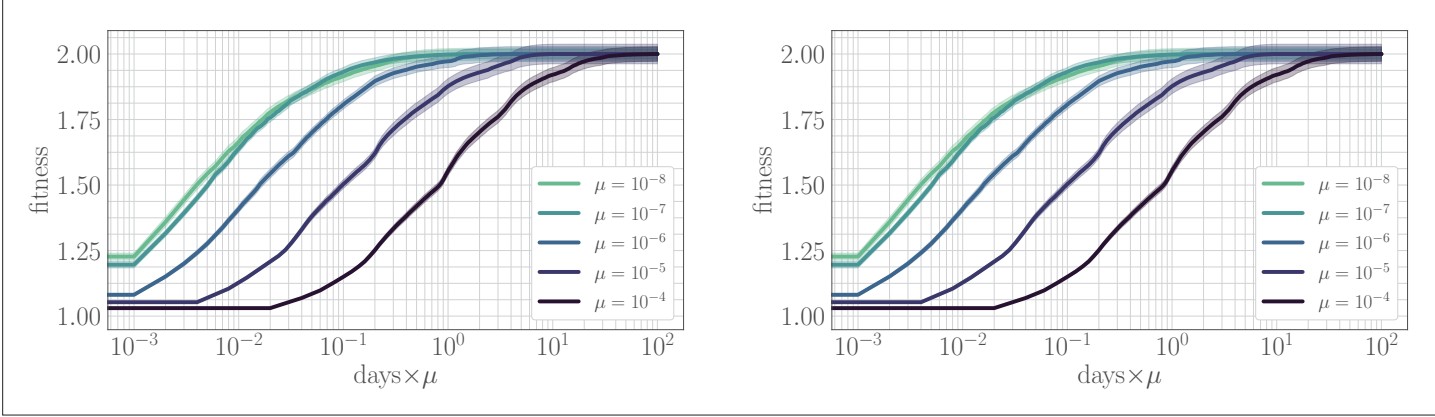

**Appendix 1—figure 1.** The effect of clonal interference on the fitness trajectory: re-scaling by μ. Left: epistatic. Right: non-epistatic. When measuring time in units of the mutation rate, clonal interference slows the fitness trajectory due to a supresion of fixation probabilities. All trajectories are initialized at one, consistent with previous result: logarithmic time axis hides the very early time dynamics.

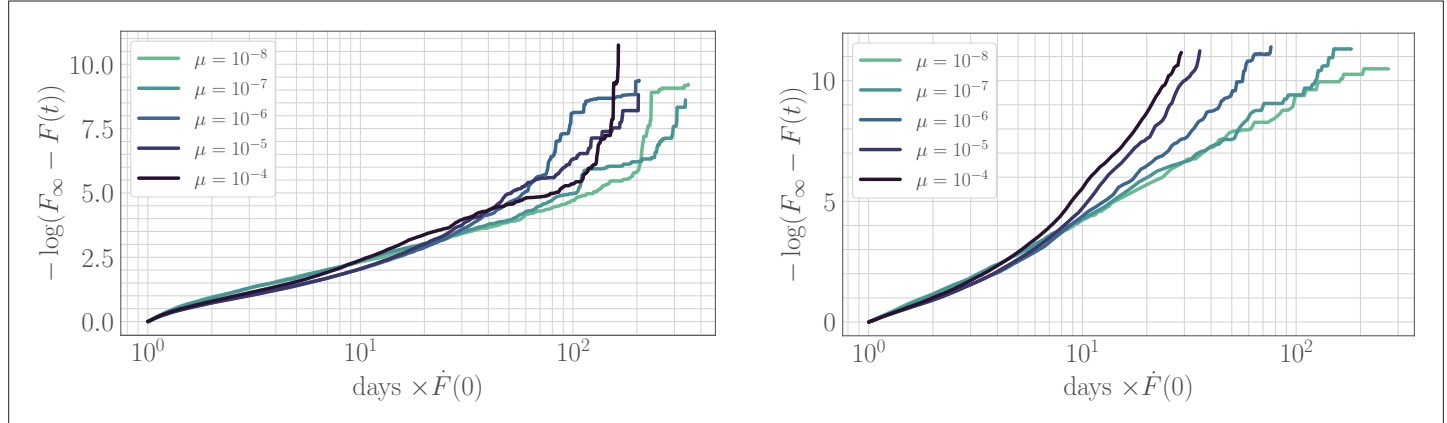

**Appendix 1—figure 2.** The effect of clonal interference on the fitness trajectory: rescaling by $\dot{F}(0)$. Left: epistatic. Right: non-epistatic. Geometrically, rescaling time by the initial rate of fitness increase ensures that slower trajectories reach the fitness peak at a later time, in qualitative agreement with the results of power-law fitting. Assuming a power law functional form as in the main text, at long times $-\log(F_\infty - F(t)) = c\log(1 + at) - \log(F_\infty - F_0)$ becomes linear with slope given by the exponent $c$. The asymptotic slope increases with μ without microscopic epistasis (left, $\beta = 0.25$), but does not clearly depend on the mutation rate with microscopic epistasis (right, $\beta = 0$).

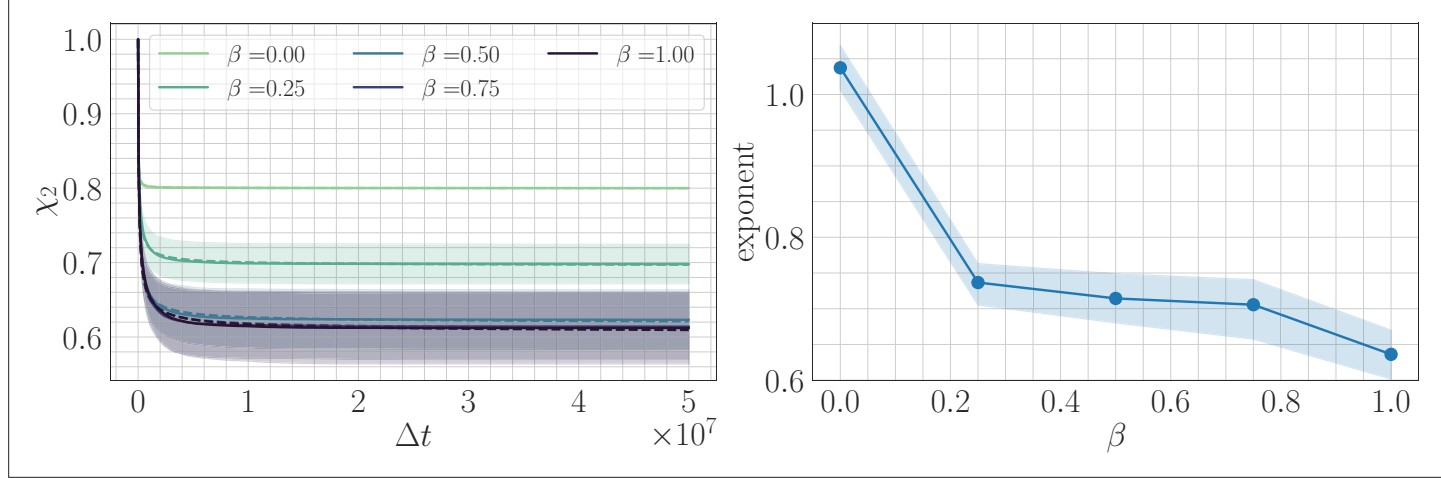

**Appendix 1—figure 3.** Two point correlation function. (Left) $\chi_2$ trajectories as a function of β and the lag time $\Delta t$ in the SSWM limit. Solid lines show the mean over 200 independent trajectories and errorbars show ± one standard deviation. Dashed line shows best fit to the power law $\chi_2(\Delta t) = \chi_2^\infty + (1 - \chi_2^\infty)/(1 + at)^c$. (Right) Relaxation exponents $c$ for $\chi_2$ as a function of β. Similar to the fitness trajectory, the exponent decreases with increasing β. Solid point shows median of the bootstrap distribution computed over 250 estimates; errorbars show 95% confidence intervals.

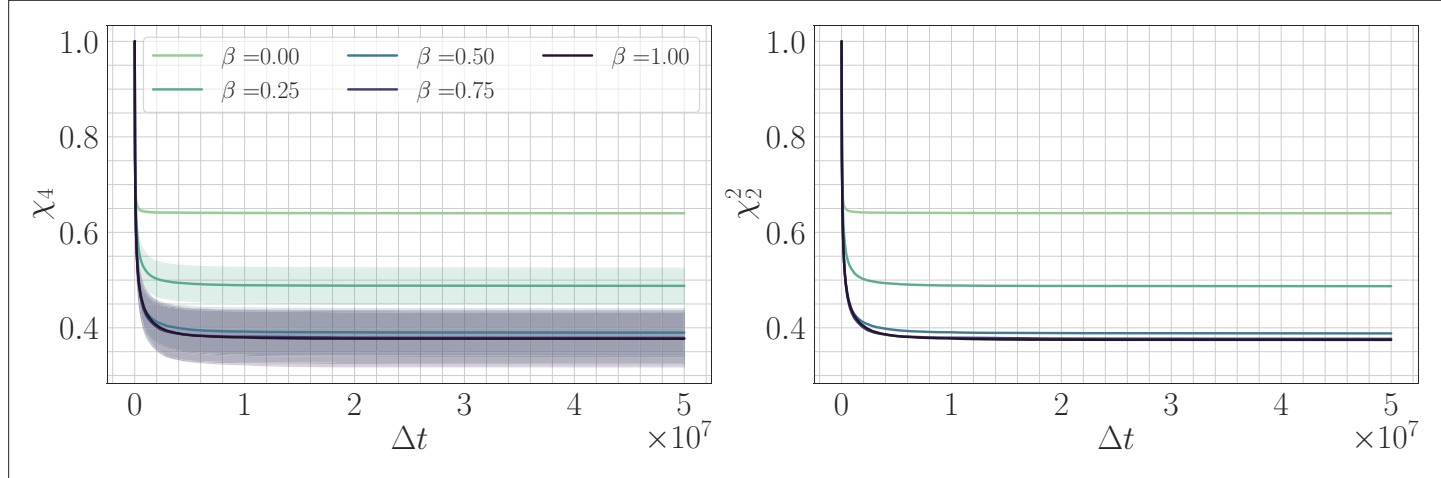

**Appendix 1—figure 4.** Four-point correlation function. (Left) $\chi_4$ trajectories as a function of β and the lag time $\Delta t$ in the SSWM limit. Solid lines show the mean over 200 independent trajectories and errorbars show ± one standard deviation. (Right) $\chi_2^2$ trajectories as a function of β. Because the genomic average of $\frac{1}{L}\sum_{i=1}^{L}\alpha_i(0)\alpha_i(\Delta t)$ is approximately equal to an average over trajectories, we find that $\chi_4(\Delta t) \approx \chi_2^2(\Delta t)$. Visually, the mean trajectories are nearly indistinguishable.

