## [Editor Report · eLife assessment]

This **important** study describes a high performance computational approach to interrogate how microscopic epistasis and clonal interference affect evolutionary dynamics in a spin glass model of microbial evolution. The study offers several insights that can aid in our understanding of the forces that operate in adaptive evolution. The evidence provided is **compelling**, with its rigorous use of models and analytical descriptions of how these forces manifest in evolution.

---

## [Referee Report · Reviewer #1 (Public Review)]

This paper presents extensive numerical simulations using a model that incorporates up to second-order epistasis to study the joint effects of microscopic epistasis and clonal interference on the evolutionary dynamics of a microbial population. Previous works that explicitly modeled microscopic epistasis typically assumed strong selection & weak mutation (SSWM), a condition that is generally not met in real-life evolutionary processes. Alternatively, another class of models coarse-grained the effects of microscopic epistasis into a generic distribution of fitness effects. The framework introduced in this paper represents an important advance with respect to these previous approaches, allowing for the explicit modeling of microscopic epistasis in non-SSWM scenarios. The modeling framework presented promises to be a valuable tool to study microbial evolution in silico.

---

## [Referee Report · Reviewer #2 (Public Review)]

This paper presents an extensive numerical study of microbial evolution using a model of fitness inspired by spin glass physics. It places special emphasis on elucidating the combined effects of microscopic epistasis, which dictates how the fitness effect of a mutation depends on the genetic background on which it occurs, and clonal interference, which describes the proliferation of and competition between multiple strains. Both microscopic epistasis and clonal interference have been observed in microbial evolution experiments, and are chief contributors to the complexity of evolutionary dynamics. Correlations between random mutations and nonlinearities associated with interactions between sub-populations consisting of competing strains make it extremely challenging to make quantitative theoretical predictions for evolutionary dynamics and associated observables such as the mean fitness. While the body of theoretical and computational research on modeling evolutionary dynamics is extensive, most theoretical efforts rely on making simplifications such as the strong selection weak mutation (SSWM) limit, which neglects clonal interference, or assumptions about the distribution of fitness effects that are not experimentally verifiable.

The authors have addressed this challenge by running a numerical microbial evolution experiment over realistic population sizes (~ 100 million cells) and timescales (~ 10,000 generations) using a spin glass model of fitness that considers pairwise interactions between mutations on distinct genetic loci. By independently tuning mutation rate as well as the strength of epistasis, the authors have shown that epistasis generically slows down the growth of fitness trajectories regardless of the amount of clonal interference. On the other hand, in the absence of epistasis, clonal interference speeds up the growth of fitness trajectories, but leaves the growth unchanged in the presence of epistasis. The authors quantitatively characterize these observations using asymptotic power law fits to the mean fitness trajectories. Further, the authors employ more simplified macroscopic models that are informed by their empirical findings, to reveal the mechanistic origins of the epistasis mediated slowing down of fitness growth. Specifically, they show that epistasis leads to a broadening of the distribution of fitness increments, leading to the fixation of a large number of mutations that confer small benefits. Effectively, this leads to an increase in the number of fixed mutations required to climb the fitness peak. This increased number of required beneficial mutations together with the decreasing availability of beneficial mutations at high fitness lead to the slowdown of fitness growth. The authors' data analysis is quite solid and their conclusions are well supported by quantitative macroscopic models. The paper also includes an interesting analysis of dynamical correlations between mutations, using tools developed in the spin glass literature.

One of the highlights of this paper is the author's astute choice of model, which strikes an impressive balance between complexity, flexibility, and numerical accessibility. In particular, the authors were able to achieve results over realistic population sizes and timescales largely because of the amenability of the model to the implementation of an efficient simulation algorithm. At the same time, the strength of epistasis and clonal interference can be tuned in a facile manner, enabling the authors to map out a phase diagram spanning these two axes. One could argue that the numerical scheme employed here would only work for a specific class of models, and is therefore not generalizable to all models of evolutionary dynamics. While this is likely true, the model is capable of recapitulating several complex aspects of microbial evolution, and is therefore not unduly restrictive.

Spin glass physics has already provided significant insights into a wide range of topics in the life sciences including protein folding, neuroscience, ecology and evolution. The present work carries this approach forward, with immediate implications for microbial evolution, and potential implications in related areas of research such as microbial ecology. In addition to the theoretical value of spin glass physics, the high performance algorithm developed in this work lays the foundation for formulating data driven approaches aimed at understanding evolutionary dynamics. In the future, there is considerable scope for utilizing data generated by such models to train machine learning algorithms for quantifying parameters associated with epistasis, clonal interference, and the distribution of fitness effects in laboratory experiments.

---

## [Author Response]

The following is the authors’ response to the original reviews.

**Reviewer 1**

We thank the reviewer for their thoughtful comments. We have addressed them below, and we believe that have significantly strengthened the clarity of the manuscript.

Main CommentsIn Fig. 2C-D, I am not sure I understand why ≈ 100 mutations fix with β = 0. In the absence of epistasis, and since the coefficients hi are sampled from a symmetric distribution centered at zero, it is to be expected that roughly half of the mutations will have positive fitness effects and thus will eventually fix in the population. With L = 250, I would have expected to see the number of fixed mutations approach ≈ 125 for β = 0. Perhaps I am missing something?

• In our simulations, we initialize all populations from a state where there are only 100 available beneficial mutations (i.e., the initial rank is always 100). Without epistasis, these initial beneficial mutations are the only beneficial mutations that will be present throughout the entire trajectory. Hence, for β = 0, only 100 beneficial mutations can fix. Previously, this information could be found in the “Materials and methods” section of the SI. To make this aspect of our simulation more clear in the revision, we have added a discussion of the initial rank to the “Landscape structure” subsection of the model definition section. In addition, we have merged “Materials and methods” with “Further simulation details” in the SI into one section, and have listed the values for the simulation parameters in the model definition section.

Along these lines, the authors show that increasing β leads to a higher number of fixed mutations. I am not sure I understand their explanation for this. In line 209 they write that as β increases, “mutations are needed to cease adaptation”. The way I see it, in the absence of epistasis the fitness peak should correspond to a genotype with ≈ L/2 mutations (the genotype carrying all mutations with hi > 0). Increasing the magnitude of microscopic epistasis (i.e., increasing β ), and assuming that there is no bias towards positive epistasis (which there shouldn’t be based on the model formulation, i.e., section "Disorder statistics" on page 4), can change the “location” of the fitness peak, such that it now corresponds to a different genotype. Statistically speaking, however, there are more genotypes with L/2 mutations than with any other number of mutations, so I would have expected that, on average, the number of mutations fixed in the population would still have been ≈ L/2 (naturally with somewhat large variation across replicates, as seems to be the case).

• With epistasis, the situation becomes more complex. The structure of our model imposes significant sign epistasis in general (i.e. mutations can be beneficial on one background genotype and deleterious on another). This means that in the presence of epistasis, more than 100 mutations can be required to reach a local optimum even when the initial rank was 100. Intuitively, this occurs because mutations that were deleterious on the ancestral background genotype can become beneficial on future genotypes. We find that this occurs consistently throughout adaptation, leading to the accumulation of more mutations with increasing epistasis.

• Please note that we use the value L = 1000 in our simulations. We have also made the fact that we use L = 1000 more clear by moving the description of the simulation parameters to the main text.

I do see how, in the clonal interference regime, there can be multiple genotypes in the population at a given time (each with a different mutational load), thus making the number of fixed mutations larger than L/2 when aggregating over all genotypes in the population. But this observation makes less intuitive sense to me in the SSWM regime. In lines 207-208, the authors state that “as beta increases, a greater number of new available beneficial mutations are generated per each typical fixation event”. While this is true, it is also the case that a greater number of mutations that would have been beneficial in the absence of epistasis are now deleterious due to negative epistasis (if I am understanding what the authors mean correctly).

• The reviewer is correct to note that in the strong clonal interference regime, there will be more accumulated mutations across the entire population than in any single strain. However, we report the number mutations that have fixed, i.e., become present in the entire population.

• We find that the typical decrease in rank (per fixation event) of the population decreases with increasing epistasis — i.e., the number of available beneficial mutations that are “consumed” when a mutation fixes is typically lower in systems with stronger epistasis.

Similarly, I am not sure I understand how one goes from equation (6) to equation (7). In particular, it would seem to me that the term 4αiαj Ji j in equation (6) should be equally likely to be positive or negative (again assuming no bias towards positive Ji j). I thus do not see why ηi j in equation (7) is sampled from a normal distribution with mean µβ instead of just mean zero.

• The reviewer is correct that, for a uniformly random initial state, αi , αj , and Ji j will be uncorrelated so that the distribution of 4αiαj Ji j can be computed exactly (and has mean zero). However, we initialize from a state with rank 100, so that we need to compute the distribution of the random variable E[αiαj Ji j|αiαj Ji j > 0, R = 100]. This is mathematically very challenging, because there are nontrivial correlations between spins even at initialization. For these reasons, we found the uniformly random approximation insufficient. This is described in the paragraph following Equation (7) in the resubmission.

Minor CommentsThe authors use a model including terms up to second-order epistasis. To be clear, I think this choice is entirely justified: as they mention in their manuscript, this structure allows to approximate any fitness model defined on a Boolean hypercube. As I understand it, the reason for not incorporating higher-order terms (as in e.g. Reddy and Desai, eLife 2021) has to do with computational efficiency, i.e., accommodating higher-order terms in equation(10) may lead to a substantial increase in computation time. Is this the case?

• The author is correct that the incorporation of higher-order terms leads to significantly more expensive computation. It’s an interesting direction of future inquiry to see if our adaptive fast fitness computation method can be extended to higher-order interactions.

**Reviewer 2**

We would like to thank the reviewer for their careful reading and their useful comments connecting our work to spin glass physics. We believe the resulting additions to the paper have made our contributions stronger, and that they reveal some novel connections between the substitution trajectory and correlation functions in spin glasses. A summary of our investigation is provided below, and we have added two paragraphs to the discussion section under the heading “Connections to spin glass physics”.

Main CommentsIn spin glasses, slowdown of dynamics could have contributions from stretched exponential relaxation of spin correlations as well as aging, each of which are associated with their own exponents. In the present model, these processes could be quantified by computing two-point correlations associated with genomic overlap, as a function of lag time as well as waiting time (generation number). The population dynamics of competing strains makes the analysis more complicated. But it should be possible to define these correlations by separately averaging over lineages starting from a single parent genome, and over distinct parent genomes. It would be interesting to see how exponents associated with these correlations relate to the exponent c associated with asymptotic fitness growth.

• To investigate this point, we first considered the two-point correlation function 〈αi (tw)αi (tw+∆t)〉 for waiting time tw and lag time ∆t. Because all spins are statistically identical, it is natural to average this over the spin index i, leading to the quantityχ2(tw,Δt)=1L⟨α(tw)⋅α(tw+Δt)⟩

Viewed as a function of ∆t for any fixed tw, it is clear that χ2(tw,0)=1L∑i=1L⟨αi(tw)2⟩=1. If m mutations with respect to α(tw) have fixed at time tw + ∆t, a similar calculation shows that χ2(tw,Δt)=1−2mL. Surprisingly, this simple derivation reveals that the two-spin correlation function commonly studied in spin glass physics is an affine transformation of the substitution trajectory commonly studied in population genetics. Moreover, it shows that the effect of tw is to change the definition of the ancestral strain, so that we may set tw = 0 without loss of generality and study the correlation function χ2(t) = 1 − 2m(t) where m(t) is the mean substitution trajectory of the population. Much of our analysis proceeds by analyzing the effect of epistasis on the accumulation of mutations. This relation provides a novel connection between this analysis and the analysis of correlation functions in the spin glass literature.

• It is well known that in the SSWM limit without epistasis, the substitution trajectory follows a power law similar to the fitness trajectory with relaxation exponent 1.0 [1]. Informed by this identity, we performed simulations in the SSWM limit and fit power laws to the correlation function χ2 as a function of time. We have verified that χ2(t) obeys a power- law relaxation with exponent roughly 1.0 for β = 0; moreover, as anticipated by the reviewer, the corresponding exponent decreases with increasing β . Nevertheless, we find that these relaxation exponents are distinct from those found for the fitness trajectory, despite following the same qualitative trend. This point is particularly interesting, as it highlights that the dynamics of fixation induce a distinct functional form at the level of the correlation functions when compared to, for example, the Glauber dynamics in statistical physics.

The strength of dynamic correlations in spin glasses can be characterized by the four-point susceptibility, which contains information about correlated spin flips. These correlations are maximized over characteristic timescales. In the context of evolution, such analysis may provide insights on the correlated accumulation of mutations on different sets of loci over different timescales. It would be interesting to see how these correlations change as a function of the mutation rate as well as the strength of epistasis.

• To study this point, we considered the four-point correlation functionχ4(t)=1L2∑i,j⟨αl(0)αl(t)αj(0)αj(t)⟩

Because spins are statistically identical, we found numerically that the genotype average is roughly equivalent to the angular average over trajectories. Inter-changing the order of

the summation and the angular averaging, we then find thatχ4(t)=⟨(1L∑tαi(0)αl(t))(1L∑jαj(0)αj(t))⟩≈χ22(t)

so that the information contained in the four-point correlation function is the same as the information contained in the two-point correlation function.

Fig. 2E and Fig. 5 together suggests an intriguing possibility when interpreted in the spin glass context. It is clear that in the absence of epistasis, clonal interference accelerates fitness growth. Fig. 2E additionally suggests that this scenario will continue to hold even in the presence of weak, but finite epistasis, but disappears for sufficiently strong epistasis. I wonder if the two regimes are separated by a phase transition at some non-trivial strength of epistasis. Indeed, the qualitative behavior appears to change from that of a random field Ising spin glass for small β , to that of a zero field Sherrington-Kirkpatrick spin glass for sufficiently large β . While the foregoing comments are somewhat speculative, perhaps a discussion along these lines, and what it means in the context of evolution could be a useful addition to the discussion section of the paper.

• We thank the reviewer for this interesting suggestion, and we have added a discussion of this point to the text in the future directions section, lines 483–489.

Minor Coments1. In the abstract (line 17-18), I recommend use of the phrase "a simulated evolving population" to avoid a possible misinterpretation of the work as experimental as opposed to numerical.

• We have added the word “simulated”.

1. In line 70, the word "the" before "statistical physics" is redundant.

• We have removed “the”.

1. To make the message in lines 294-295 visually clear, I recommend keeping the Y-axis scale bars constant across Fig. 4A and Fig. 4B.

• We appreciate the suggestion. However, we found that when putting the two figures on the same scale, because the agreement is only qualitative and not quantitative (as emphasized in the text), it becomes difficult to view the trend in both systems. For this reason, we have chosen to keep the figure as-is.

1. Fig. 6 caption states: "Without epistasis, the rank decreases with increasing µ". It should be "rank increases".

• We have fixed this.

1. In the last sentence in the caption to Fig. 8, the labels "(A, β = 0)" and "(B, β = 0.25)" need to be swapped.

• We have fixed this.

**Editor Comments**

We thank the editor for pointing our attention towards these three interesting references, in particular the second, which appears most relevant to our work. We have added a discussion of reference 2 in the future directions section (lines 471–482), commenting on how to determine the contribution of within-path clonal interference to the fitness dynamics in our model. We have also added a reference to article 3 in the model description, commenting on the importance of sign epistasis and the prevalence of sign epistasis in our model with β > 0.

References

1. Good BH, Desai MM. The impact of macroscopic epistasis on long-term evolutionary dynamics. Genetics. 2015.